# WOULD DECENTRALIZATION HURT GENERALIZATION?

## ABSTRACT

Decentralized stochastic gradient descent (D-SGD) allows collaborative learning on massive devices without the control of a central server. Existing theory suggests that decentralization degrades generalizability, which conflicts with experimental results in large-batch settings that D-SGD generalizes better than centralized SGD (C-SGD). This work presents a new theory that reconciles the conflict between the two perspectives. We prove that D-SGD introduces an implicit regularization that simultaneously penalizes (1) the sharpness of the learned minima and (2) the consensus distance between the global averaged model and local models. We then prove that the implicit regularization is amplified in large-batch settings when the linear scaling rule is applied. We further analyze the escaping efficiency of D-SGD and show that D-SGD favors super-quadratic flat minima. Experiments are in full agreement with our theory. The code will be released publicly. To our best knowledge, this is the first work on the implicit regularization and escaping efficiency of D-SGD.

## 1 INTRODUCTION

Decentralized stochastic gradient descent (D-SGD) enables simultaneous model training on massive workers without being controlled by a central server, where every worker communicates only with its directly connected neighbors (Xiao & Boyd, 2004; Lopes & Sayed, 2008; Nedic & Ozdaglar, 2009; Lian et al., 2017; Koloskova et al., 2020). This decentralization avoids the requirements of a costly central server with heavy communication and computation burdens. Despite the absence of a central server, existing theoretical results demonstrate that the massive models on the edge converge to a unique steady consensus model (Shi et al., 2015; Lian et al., 2017; Lu et al., 2011), with asymptotic linear speedup in convergence rate (Lian et al., 2017) as the distributed centralized SGD (C-SGD) does (Dean et al., 2012; Li et al., 2014). Consequently, D-SGD offers a promising distributed learning solution with significant advantages in privacy (Nedic, 2020), scalability (Lian et al., 2017), and communication efficiency (Ying et al., 2021b).

However, existing theoretical studies show that the decentralization nature of D-SGD introduces an additional positive term into the generalization error bounds, which suggests that decentralization may hurt generalization (Sun et al., 2021; Zhu et al., 2022). This poses a crippling conflict with empirical results by Zhang et al. (2021) which show that D-SGD generalizes better than C-SGD by a large margin in large batch settings; see Figure 1. This conflict signifies that the major characteristics were overlooked in the existing literature. Therefore,

> *would decentralization hurt generalization?*

This work reconciles the conflict. We prove that decentralization introduces implicit regularization in D-SGD, which promotes the generalization. To our best knowledge, this is the first paper that surprisingly shows the advantages of D-SGD in generalizability, which redresses the former misunderstanding. Specifically, our contributions are in twofold.

- We prove that the mean iterate of D-SGD closely follows the path of C-SGD on a regularized loss, which is the addition of the original loss and a regularization term introduced by decentralization. This regularization term penalizes the largest eigenvalue of the Hessian matrix, as well as the consensus distance (see Theorem 1). These regularization effects are shown to be considerably amplified in large-batch settings (see Theorem 2), which is consistent with our visualization (see

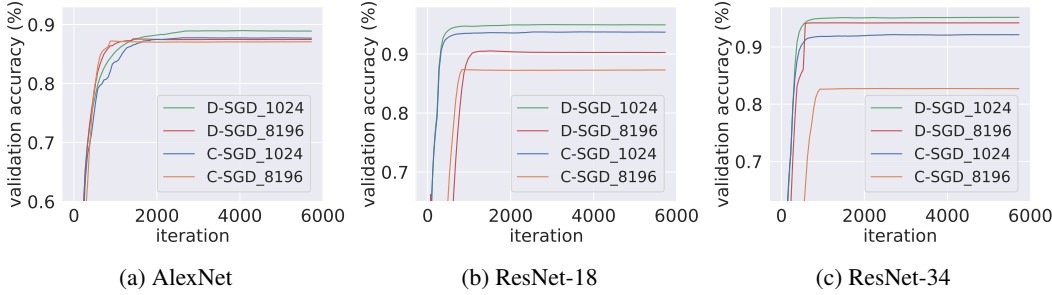

Figure 1: Comparison of the validation accuracy of C-SGD and D-SGD on CIFAR-10. The number of workers (one GPU as a worker) is set as 16; and the local batch size is set as 64, and 512 per worker (1024 and 8196 total batch size). The training setting is included in Section 5.

Figure 4) and the empirical results in (Zhang et al., 2021). To prove the above results, we apply second-order multivariate Taylor approximation (Königsberger, 2013) on the gradient diversity (see Equation (5)) to derive the regularized loss. Then, we prove that the regularization term contained in the regularized loss scales positively with the largest Hessian eigenvalue, which suggests that D-SGD implicitly minimizes the sharpness of the learned minima (see Lemma C.2).

- We prove the first result on the expected escaping speed of D-SGD from local minima (see Theorem 3). Our results show that D-SGD prefers super-quadratic flat minima to sub-quadratic minima with higher probability (see Proposition 4). The proof is based on the construction of a stochastic differential equation (SDE) approximation (Jastrzebski et al., 2017; M et al., 2017; Li et al., 2021) of D-SGD.

## 2 RELATED WORK

**Flatness and generalization.** The flatness of minimum is a commonly used concept in the optimization and machine learning literature and has long been regarded as a proxy of generalization (Hochreiter & Schmidhuber, 1997; Izmailov et al., 2018; Jiang et al., 2020). Intuitively, the loss around a flat minimum varies slowly in a large neighborhood, while a sharp minimum increases rapidly in a small neighborhood (Hochreiter & Schmidhuber, 1997). Through the lens of the minimum description length theory (Rissanen, 1983), flat minimizers tend to generalize better than sharp minimizers, since they are specified with lower precision (Keskar et al., 2017). From a Bayesian perspective, sharp minimizers have posterior distributions highly concentrated around them, indicating that they are more specialized on the training set and thus are less robust to data perturbations than flat minimizers (MacKay, 1992; Chaudhari et al., 2019).

**Generalization of large-batch training.** Large-batch training is of significant interest for deep learning deployment, which can contribute to a significant speed-up in training neural networks (Goyal et al., 2017; You et al., 2018; Shallue et al., 2019). Unfortunately, it is widely observed that in the centralized learning setting, large-batch training often suffers from a drastic generalization degradation, even with fine-tuned hyper-parameters, from both empirical (Chen & Huo, 2016; Keskar et al., 2017; Hoffer et al., 2017; Shallue et al., 2019; Smith et al., 2020) and theoretical (Li et al., 2021) aspects. An explanation of this phenomenon is that large-batch training leads to "sharper" minima (Keskar et al., 2017), which are more sensitive to perturbations (Hochreiter & Schmidhuber, 1997).

**Development of D-SGD.** The earliest work of classical decentralized optimization can be traced back to Tsitsiklis (1984), Tsitsiklis et al. (1986) and Nedic & Ozdaglar (2009). D-SGD, a typical decentralized optimization algorithm, has been extended to various settings in deep learning, including time-varying topologies (Lu & Wu, 2020; Koloskova et al., 2020), asynchronous settings (Lian et al., 2018; Xu et al., 2021; Nadiradze et al., 2021), directed topologies (Assran et al., 2019; Taheri et al., 2020), and data-heterogeneous scenarios (Tang et al., 2018; Vogels et al., 2021).

**Generalization of D-SGD.** Recently, Sun et al. (2021) and Zhu et al. (2022) have established generalization bounds of D-SGD and have shown that decentralized training hurts generalization.

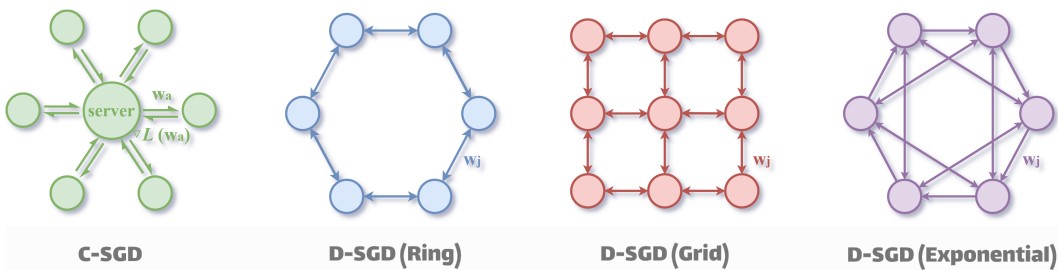

Figure 2: An illustration of C-SGD and D-SGD.

However, these works do not analyze the sharpness reduction effect of D-SGD and cannot explain why D-SGD can generalize better than C-SGD in large batch settings. Another work by Zhang et al. (2021) demonstrates that D-SGD introduces an "additional" landscape-dependent noise, which improves the convergence of D-SGD. However, the direction, magnitude, and shape of the noise remain unexplored. In contrast, we rigorously prove that the additional noise of D-SGD (i.e., the gradient diversity in Equation (4)) biases the trajectory of D-SGD towards flatter minima, which may play a distinct role in shaping the generalizability of D-SGD.

## 3 PRELIMINARIES

Suppose that $\mathcal{X} \subseteq \mathbb{R}^{d_x}$ and $\mathcal{Y} \subseteq \mathbb{R}$ are the input and output spaces, respectively. We denote the training set as $\mu = \{z_1, \ldots, z_N\}$, where $z_\zeta = (x_\zeta, y_\zeta), \zeta = 1, \ldots, N$ are sampled independent and identically distributed (i.i.d.) from an unknown data distribution $\mathcal{D}$ defined on $\mathcal{Z} = \mathcal{X} \times \mathcal{Y}$. The goal of supervised learning is to learn a predictor (hypothesis) $g(\cdot; \mathbf{w})$, parameterized by $\mathbf{w} = \mathbf{w}(z_1, z_2, \ldots, z_N) \in \mathbb{R}^d$, to approximate the mapping between the input variable $x \in \mathcal{X}$ and the output variable $y \in \mathcal{Y}$, based on the training set $\mu$. Let $c : \mathcal{Y} \times \mathcal{Y} \mapsto \mathbb{R}^+$ be a function that evaluates the prediction performance of hypothesis $g$. The loss of a hypothesis $g$ with respect to (w.r.t.) the example $z_\zeta = (x_\zeta, y_\zeta)$ is denoted by $\boldsymbol{L}(\mathbf{w}; z_\zeta) = c(g(x_\zeta; \mathbf{w}), y_\zeta)$, which measures the effectiveness of the learned model. Then, the empirical and population risks of $\mathbf{w}$ are defined as follows:

$$\boldsymbol{L}^\mu(\mathbf{w}) = \frac{1}{N} \sum_{\zeta=1}^N \boldsymbol{L}(\mathbf{w}; z_\zeta), \quad \boldsymbol{L}(\mathbf{w}) = \mathbb{E}_{z \sim D}[\boldsymbol{L}(\mathbf{w}; z)].$$

**Distributed learning.** Distributed learning jointly trains a learning model $\mathbf{w}$ on multiple workers (Shamir & Srebro, 2014). In this framework, the $j$-th worker ($j = 1, \ldots, m$) can access $|\mu_j|$ independent and identically distributed (i.i.d.) training examples $\mu_j = \{z_{j,1}, \ldots, z_{j,|\mu_j|}\}$, drawn from the data distribution $\mathcal{D}$. In this case, the global empirical risk of $\mathbf{w}$ is

$$\boldsymbol{L}^\mu(\mathbf{w}) = \frac{1}{m} \sum_{j=1}^m \boldsymbol{L}^{\mu_j}(\mathbf{w}),$$

where $\boldsymbol{L}^{\mu_j}(\mathbf{w}) = \frac{1}{|\mu_j|} \sum_{\zeta=1}^{|\mu_j|} \boldsymbol{L}(\mathbf{w}; z_{j,\zeta})$ denotes the local empirical risk on the $j$-th worker and $z_{j,\zeta} \in \mu_j$ ($\zeta = 1, \ldots, |\mu_j|$) stands for the local training data.

**Distributed centralized stochastic gradient descent (C-SGD).**[1] In C-SGD, there is only one centralized model $\mathbf{w}_a(t)$. C-SGD (Dean et al., 2012; Li et al., 2014) updates the model by

$$\mathbf{w}_a(t+1) = \mathbf{w}_a(t) - \frac{1}{m} \sum_{j=1}^m \eta \cdot \overbrace{\nabla \boldsymbol{L}^{\mu_j(t)}(\mathbf{w}_a(t))}^{\text{Local gradient computation}}, \tag{1}$$

where $\eta$ denotes the learning rate, $\mu_j(t) = \{z_{j,1}, \ldots, z_{j,|\mu_j(t)|}\}$ denotes the local training batch independent and identically distributed (i.i.d.) drawn from the data distribution $\mathcal{D}$ at the $t$-th iteration,

---

[1]The word "centralized" indicates that in C-SGD, there is a central server receiving gradient information from local workers (see Figure 2).

and $\nabla \boldsymbol{L}^{\mu_j(t)}(\mathbf{w}) = \frac{1}{|\mu_j(t)|} \sum_{\zeta(t)=1}^{|\mu_j(t)|} \nabla \boldsymbol{L}(\mathbf{w}; z_{j,\zeta(t)})$ stands for the local mini-batch gradient of $\boldsymbol{L}$ w.r.t. the first argument $\mathbf{w}$. The total batch size of C-SGD at $t$-th iteration is $|\mu(t)| = \sum_{j=1}^{m} |\mu_j(t)|$. In the next section, we will show that C-SGD equals the single-worker SGD with a larger batch size.

**Decentralized stochastic gradient descent (D-SGD).** The goal of D-SGD is to learn a consensus model $\mathbf{w}_a(t) = \frac{1}{m} \sum_{j=1}^{m} \mathbf{w}_j(t)$ on $m$ workers, where $\mathbf{w}_j(t)$ stands for the $d$-dimensional local model on the $j$-th worker. We denote $\mathbf{P} = [\mathbf{P}_{j,k}] \in \mathbb{R}^{m \times m}$ as a doubly stochastic gossip matrix (see Definition A.1) that characterizes the underlying topology $\mathcal{G}$. The vanilla Adapt-While-Communicate (AWC) version of the mini-batch D-SGD (Nedic & Ozdaglar, 2009; Lian et al., 2017) updates the model on the $j$-th worker by

$$\mathbf{w}_j(t+1) = \overbrace{\sum_{j=1}^{m} \mathbf{P}_{j,k} \mathbf{w}_k(t)}^{\text{Communication}} - \eta \cdot \overbrace{\nabla \boldsymbol{L}^{\mu_j(t)}(\mathbf{w}_j(t))}^{\text{Local gradient computation}} . \tag{2}$$

For a more detailed background of D-SGD, please refer to Appendix A .

## 4 THEORETICAL RESULTS

This section shows the implicit regularization effect and the escaping efficiency of D-SGD. We start by showing that D-SGD can be interpreted as C-SGD on a regularized loss. Then we prove that the regularization term in the new loss scales positively with the largest Hessian eigenvalue (see Theorem 1), which suggests that D-SGD implicitly minimizes the sharpness. Next, we prove that the regularization effect will increase with the total batch size if we apply the linear scaling rule (see Theorem 2), which justifies the superiority of D-SGD in large-batch settings. Finally, we prove the escaping efficiency of D-SGD beyond the quadratic assumption (see Theorem 3) and show that D-SGD favors super-quadratic minima (see Proposition 4).

### 4.1 D-SGD IS EQUIVALENT TO C-SGD ON A REGULARIZED LOSS

In this subsection, we theoretically compare D-SGD and C-SGD. We prove that D-SGD is equivalent to C-SGD on regularized loss with an extra positive regularization term, as shown in the following theorem.

**Theorem 1** (Implicit regularization of D-SGD). *Given that the loss $\boldsymbol{L}$ is continuous and has fourth-order partial derivatives, denote the weight diversity matrix as $\boldsymbol{\Xi}(t) = \frac{1}{m} \sum_{j=1}^{m} (\mathbf{w}_j(t) - \mathbf{w}_a(t))(\mathbf{w}_j(t) - \mathbf{w}_a(t))^T$, its diagonal matrix as $\boldsymbol{\Xi}^*(t)$, and the $d$-dimensional all-ones vector as $\mathbf{1}$. With a probability greater than $1 - \mathcal{O}(\eta)$, the mean iterate of D-SGD becomes*

$$\mathbb{E}_{\substack{\mu_j(t) \sim D \\ j=1,\ldots,m}} [\mathbf{w}_a(t+1)]$$

$$= \mathbf{w}_a(t) - \eta \nabla \underbrace{\left[ \boldsymbol{L}(\mathbf{w}_a(t)) + \frac{1}{2} \mathrm{Tr}(\boldsymbol{H}(\mathbf{w}_a(t))\boldsymbol{\Xi}^*(t)) \right]}_{\text{the regularized loss}} + \mathcal{O}(\eta^2 \mathbf{1}) + \mathcal{O}\left(\eta \|\mathbf{w}_j(t) - \mathbf{w}_a(t)\|_2^3 \mathbf{1}\right), \tag{3}$$

*Under mild assumptions in Lemma C.2, D-SGD implicitly regularizes*

$$reg(\underset{j=1,\ldots,m}{\mathbf{w}_j(t)}) = \underbrace{\lambda_{\boldsymbol{H}(\mathbf{w}_a(t)),1}}_{\text{maximum Hessian eigenvalue}} \cdot \underbrace{\mathrm{Tr}(\boldsymbol{\Xi}(t))}_{\text{consensus distance}} .$$

The first term $\lambda_{\boldsymbol{H}(\mathbf{w}_a(t)),1}$ is commonly regraded as a sharpness measure (Jastrzebski et al., 2017; Wen et al., 2020). It is related to the $(\mathcal{C}_\epsilon, A)$-sharpness (i.e., $\max_{\mathbf{w}' \in \mathcal{C}_\epsilon} \boldsymbol{L}(\mathbf{w} + A\mathbf{w}') - \boldsymbol{L}(\mathbf{w}))$ in Keskar et al. (2017) and is an equivalent measure to the Sharpness Aware Minimization (SAM) loss proposed by Foret et al. (2021) at a local minimum (Zhuang et al., 2022). Theorem 1 shows that the decentralization navigates D-SGD towards the flatter directions, in order to lower the regularization term $\lambda_{\boldsymbol{H}(\mathbf{w}_a(t)),1}$. The second term, the trace of $\boldsymbol{\Xi}(t)$, equals to the *consensus distance*, a key component measuring the overall effect of decentralized learning (Kong et al., 2021),

$$\text{consensus distance} = \frac{1}{m} \sum_{j=1}^{m} (\mathbf{w}_j(t) - \mathbf{w}_a(t))^T (\mathbf{w}_j(t) - \mathbf{w}_a(t)).$$

Consequently, Theorem 1 also suggests that D-SGD implicitly controls the discrepancy between the global averaged model $\mathbf{w}_a(t)$ and the local models $\mathbf{w}_j(t)$ $(j = 1, \ldots, m)$ during training.

Our derived implicit regularization on the sharpness of learned minima is similar to how label noise (Blanc et al., 2020; Damian et al., 2021) and artificial noise (Orvieto et al., 2022) smooth the loss function in centralized gradient methods, including distributed centralized gradient methods (C-SGD) and single-worker gradient methods. To the best of our knowledge, this is the first work that shows D-SGD is equivalent to C-SGD on a regularized loss with implicit sharpness regularization. In the existing literature, initial efforts have viewed D-SGD as C-SGD in a higher-dimensional space that penalizes the weight norm $\|\mathbf{W}\|_{\mathbf{I-P}}^2$, where $\mathbf{W} = [\mathbf{w}_1, \cdots, \mathbf{w}_m]^T \in \mathbb{R}^{m \times d}$ stands for all local models across the network (Yuan et al., 2021; Gurbuzbalaban et al., 2022).

We summarize the proof sketch below. The full proof is given in Appendix C.

**Proof sketch.**

**(1) Deriving the dynamics of the global averaged model** [2]**.** We first start by rewriting the update of the global averaged model $\mathbf{w}_a(t)$ of D-SGD as follows,

$$\mathbf{w}_a(t+1) = \mathbf{w}_a(t) - \eta \Big[ \underbrace{\nabla \boldsymbol{L}\left(\mathbf{w}_a(t)\right)}_{\text{unbiased gradient}} + \underbrace{\nabla \boldsymbol{L}\left(\mathbf{w}_a(t)\right) - \nabla \boldsymbol{L}^{\mu(t)}\left(\mathbf{w}_a(t)\right)}_{\text{gradient noise over the superbatch } \mu(t)}$$

$$+ \underbrace{\frac{1}{m} \sum_{j=1}^{m} [\nabla \boldsymbol{L}^{\mu_j(t)}\left(\mathbf{w}_j(t)\right) - \nabla \boldsymbol{L}^{\mu_j(t)}\left(\mathbf{w}_a(t)\right)]}_{\text{gradient diversity among workers}} \Big]. \tag{4}$$

**Remark.** The equality shows that decentralization introduces an additional noise, which characterizes the gradient diversity between the global averaged model $\mathbf{w}_a(t)$ and the local models $\mathbf{w}_j(t)$ $(j = 1, \ldots, m)$. It implies that distributed centralized SGD, which has constant zero gradient diversity, is equivalent to standard single-worker SGD with larger batch size. Note that the gradient diversity also equals to zero on quadratic loss $\boldsymbol{L}$ (see Corollary C.1). Consequently, the quadratic approximation in the analysis of mini-batch SGD (Zhu et al., 2019b; Ibayashi & Imaizumi, 2021; Liu et al., 2021) fails to capture how decentralization affects the training dynamics of D-SGD.

**(2) Performing Taylor expansion on the gradient diversity.** Analyzing the effect of the gradient diversity on the training dynamics of D-SGD on the general non-convex losses is highly non-trivial. Technically, we perform a second-order Taylor expansion on the gradient diversity around $\mathbf{w}_a(t)$, omitting the high-order residuals $R$:

$$\frac{1}{m} \sum_{j=1}^{m} [\nabla \boldsymbol{L}^{\mu_j(t)}\left(\mathbf{w}_j(t)\right) - \nabla \boldsymbol{L}^{\mu_j(t)}\left(\mathbf{w}_a(t)\right)]$$

$$= \frac{1}{m} \sum_{j=1}^{m} \boldsymbol{H}^{\mu_j(t)}(\mathbf{w}_a(t))(\mathbf{w}_j(t) - \mathbf{w}_a(t)) + \frac{1}{2m} \sum_{j=1}^{m} \boldsymbol{T}^{\mu_j(t)}(\mathbf{w}_a(t)) \otimes [(\mathbf{w}_j(t) - \mathbf{w}_a(t))(\mathbf{w}_j(t) - \mathbf{w}_a(t))^T].$$

Here $\boldsymbol{H}^{\mu_j(t)}(\mathbf{w}_a(t)) \triangleq \frac{1}{|\mu_j(t)|} \sum_{\zeta(t)=1}^{|\mu_j(t)|} \boldsymbol{H}(\mathbf{w}_a(t); z_{j,\zeta(t)})$ stands for the empirical Hessian at $\mathbf{w}_a(t)$ and $\boldsymbol{T}^{\mu_j(t)}(\mathbf{w}_a(t)) \triangleq \frac{1}{|\mu_j(t)|} \sum_{\zeta(t)=1}^{|\mu_j(t)|} \boldsymbol{T}(\mathbf{w}_a(t); z_{j,\zeta(t)})$ denotes the empirical third-order partial derivative tensor at $\mathbf{w}_a(t)$, where $\mu_j(t)$ and $z_{j,\zeta(t)}$ follows the notation in Equation (1).

Analogous to the works investigating the SGD dynamics (M et al., 2017; Zhu et al., 2019b; Ziyin et al., 2022; Wu et al., 2022), we will calculate the expectation and covariance of the gradient diversity. The expectation of gradient diversity is calculated first as follows. We defer the analysis of its covariance to Subsection 4.3. Taking expectation over all local mini-batches $\mu_j(t)$ $(j = 1, \ldots, m)$ provides[3]

$$\mathbb{E}_{\substack{\mu_j(t) \sim D \\ j=1,\ldots,m}} \Big[ \frac{1}{m} \sum_{j=1}^{m} [\nabla \boldsymbol{L}^{\mu_j(t)}\left(\mathbf{w}_j(t)\right) - \nabla \boldsymbol{L}^{\mu_j(t)}\left(\mathbf{w}_a(t)\right)] \Big]$$

---

[2]Note that there is no central server In D-SGD. In the following we analyze the training dynamics of the global averaged model $\mathbf{w}_a(t)$ of D-SGD, which has been proved to be close to the individual models $\mathbf{w}_j(t)(j = 1, ..., m)$ (Yuan et al., 2016; Fallah et al., 2022).

[3]Taking expectation over $\mu_j(t)$ means taking expectation over all $z_{j,\zeta(t)}$ $(\zeta(t) = 1, \ldots, |\mu_j|)$.

$$=\boldsymbol{H}(\mathbf{w}_a(t))\underbrace{\frac{1}{m}\sum_{j=1}^m(\mathbf{w}_j(t){-}\mathbf{w}_a(t))}_{=0}+\frac{1}{2}\boldsymbol{T}(\mathbf{w}_a(t))\otimes[\frac{1}{m}\sum_{j=1}^m(\mathbf{w}_j(t){-}\mathbf{w}_a(t))(\mathbf{w}_j(t){-}\mathbf{w}_a(t))^T]+R.$$

The $i$-th entry of the above equation will be

$$\mathbb{E}_{\substack{\mu_j(t)\sim D\\j=1,\ldots,m}}\big[\frac{1}{m}\sum_{j=1}^m\big[\partial_i\boldsymbol{L}^{\mu_j(t)}\big(\mathbf{w}_j(t)\big){-}\partial_i\boldsymbol{L}^{\mu_j(t)}\big(\mathbf{w}_a(t)\big)\big]\big]$$

$$=\underbrace{\frac{1}{2}\sum_{k,l}\partial_{ikl}^3\boldsymbol{L}(\mathbf{w}_a(t))\frac{1}{m}\sum_{j=1}^m\big(\mathbf{w}_j(t){-}\mathbf{w}_a(t)\big)_k\big(\mathbf{w}_j(t){-}\mathbf{w}_a(t)\big)_l}_{=\partial_i\sum_{kl}\partial_{kl}^2\boldsymbol{L}(z_n)\frac{1}{m}\sum_{j=1}^m(\mathbf{w}_j(t)-\mathbf{w}_a(t))_k(\mathbf{w}_j(t)-\mathbf{w}_a(t))_l}+\mathcal{O}\left(\|\mathbf{w}_j(t){-}\mathbf{w}_a(t)\|_2^3\right),\quad(5)$$

where $\big(\mathbf{w}_j(t){-}\mathbf{w}_a(t)\big)_k$ denotes the $k$-th entry of the vector $\mathbf{w}_j(t){-}\mathbf{w}_a(t)$. The equality in the brace is due to Clairaut's theorem (Rudin et al., 1976).

Then we prove that with probability greater than $1{-}\mathcal{O}(\eta)$, the iterate of D-SGD can be written as

$$\mathbb{E}_{\substack{\mu_j(t)\sim D\\j=1,\ldots,m}}\big[\mathbf{w}_a(t+1)\big]$$

$$=\mathbf{w}_a(t){-}\eta\nabla\underbrace{\left[\boldsymbol{L}\left(\mathbf{w}_a(t)\right)+\frac{1}{2}\operatorname{Tr}(\boldsymbol{H}(\mathbf{w}_a(t))\boldsymbol{\Xi}^*(t))\right]}_{\text{the regularized loss}}+\mathcal{O}(\eta^{\frac{1}{2}}\mathbf{1})+\mathcal{O}\left(\eta\|\mathbf{w}_j(t){-}\mathbf{w}_a(t)\|_2^3\mathbf{1}\right).$$

**(3) Controlling the top Hessian eigenvalue with** $\operatorname{Tr}(\boldsymbol{H}(\mathbf{w}_a(t))\boldsymbol{\Xi}^*(t))$**.** According to Lemma C.2, we obtain

$$0\leq\operatorname{Tr}(\boldsymbol{H}(\mathbf{w}_a(t))\boldsymbol{\Xi}^*(t))\leq\underbrace{\lambda_{\boldsymbol{H}(\mathbf{w}_a(t)),1}}_{\text{sharpness}}\cdot\underbrace{\operatorname{Tr}(\boldsymbol{\Xi}(t))}_{\text{consensus distance}}\leq d_1\operatorname{Tr}(\boldsymbol{H}(\mathbf{w}_a(t))\boldsymbol{\Xi}^*(t)),$$

where $\lambda_{\boldsymbol{H}(\mathbf{w}_a(t)),1}$ denotes the largest eigenvalue of $\boldsymbol{H}(\mathbf{w}_a(t))$ and $d_1$ stands for the marginal contribution of $\lambda_{\boldsymbol{H}(\mathbf{w}_a(t)),1}$ on the full spectrum of $\boldsymbol{H}(\mathbf{w}_a(t))$ (i.e., $\lambda_{\boldsymbol{H}(\mathbf{w}_a(t)),1}=\frac{d_1}{d}\operatorname{Tr}(\boldsymbol{H}(\mathbf{w}_a(t)))$). Therefore, combined with Equation (3), we conclude that D-SGD also implicitly regularizes $\lambda_{\boldsymbol{H}(\mathbf{w}_a(t)),1}\cdot\operatorname{Tr}(\boldsymbol{\Xi}(t))$.

## 4.2 AMPLIFIED REGULARIZATION OF D-SGD IN LARGE-BATCH SETTING

In practice, the decentralization (and also distribution) ordinarily implies an equivalent large total batch size, since a massive number of workers are involved in the system in many practical scenarios. Moreover, large-batch training can enhance the utilization of super computing facilities and further speeds up the entire training process. Thus, studying the large-batch setting is of significant interest for fully understanding the application of D-SGD.

Despite the importance, theoretical understanding of the generalization of large-batch training in D-SGD remains an open problem. This subsection examines how the total batch size affects the sharpness reduction effect of D-SGD if the linear scaling rule, as presented below, is applied.

**Linear scaling rule (LSR).** The linear scaling rule is a widely used hyper-parameter-free rule for deep learning (Krizhevsky, 2014; He et al., 2016a; Goyal et al., 2017; Bottou et al., 2018; Smith et al., 2020), which states that a fixed learning rate to total batch size ratio allows maintaining generalization performance when the total batch size increases.

**Theorem 2.** *Suppose that the averaged gradient norm satisfies* $\frac{1}{m}\sum_{j=1}^m\|\nabla\boldsymbol{L}\left(\mathbf{w}_j(t)\right)\|^2\leq(1{+}\frac{1-\lambda}{4})\frac{1}{m}\sum_{j=1}^m\|\nabla\boldsymbol{L}\left(\mathbf{w}_j(t{+}1)\right)\|^2$, *where* $1{-}\lambda$ *denotes the spectral gap (see Definition A.2). The sharpness regularization coefficient*[4] *of D-SGD (i.e.,* $\operatorname{Tr}(\boldsymbol{\Xi}(t))$*) at $t$-th iteration is* $\mathcal{O}(|\mu(t)|^2(1+\frac{1}{m}\sum_{j=1}^m\frac{1}{|\mu_j(t)|}))$*, which increases with the total batch size* $|\mu(t)|$ *if we apply the linear scaling rule.*

---

[4]Recall that Theorem 1 implies that the loss function D-SGD optimizes is close to the original loss $L$ plus $\frac{1}{2}\operatorname{Tr}\left(\boldsymbol{\Xi}(t)\right)\cdot\lambda_{\boldsymbol{H}(\mathbf{w}_a(t)),1}$. The second term $\lambda_{\boldsymbol{H}(\mathbf{w}_a(t)),1}$ is a sharpness measure, and the first term $\operatorname{Tr}(\boldsymbol{\Xi}(t))$ is the "regularization coefficient" which characterizes the strength of the sharpness regularization.

Theorem 2 states that the sharpness regularization effect of D-SGD is amplified in large-batch settings if we apply the linear scaling rule. It is worth noting that this amplified sharpness regularization effect requires no additional communication and computation, which verifies that significant advantages in generalizability surprisingly exist in the large-batch D-SGD. The proof is included in Appendix C.

## 4.3 ESCAPING EFFICIENCY OF D-SGD FROM LOCAL MINIMA

This subsection presents an analysis of the escaping efficiency of D-SGD, based on the construction of a stochastic differential equation (SDE) approximation (Jastrzebski et al., 2017; M et al., 2017; Li et al., 2021) of D-SGD. This escaping efficiency analysis shows that D-SGD favors super-quadratic minima.

To construct the SDE approximation of D-SGD, we combine Equation (3) and Equation (4) and write the iterates of D-SGD as follows,

$$\mathbf{w}_a(t+1)$$
$$= \mathbf{w}_a(t) - \eta \nabla \Big[ \boldsymbol{L}\left(\mathbf{w}_a(t)\right) + \frac{1}{2}\operatorname{Tr}(\boldsymbol{H}(\mathbf{w}_a(t))\boldsymbol{\Xi}^*(t)) \Big] + \eta \epsilon^0(t) + \mathcal{O}(\eta^{\frac{1}{2}}\mathbf{1}) + \mathcal{O}\left(\eta \|\mathbf{w}_j(t) - \mathbf{w}_a(t)\|_2^3 \mathbf{1}\right), \quad (6)$$

where $\epsilon^0(t)$ denotes the zero-mean noise in D-SGD. Applying Lemma C.4, Equation (6) can be viewed as the discretization of the following SDE

$$\mathrm{d}\mathbf{w}_a(t) = -\Big[\nabla \boldsymbol{L}\left(\mathbf{w}_a(t)\right) + \frac{1}{2}\boldsymbol{T}(\mathbf{w}_a(t)) \otimes \boldsymbol{\Xi}^*(t)\Big]\mathrm{d}t + \sqrt{\eta \boldsymbol{\Sigma}_{\mathrm{D}(t)}}\mathrm{d}W(t),$$

where $\otimes$ denotes the tensor product (see Appendix A.2), $\boldsymbol{\Sigma}_{\mathrm{D}(t)}$ denotes the covariance matrix of the total noise $\epsilon_{\mathrm{D}(t)} = \frac{1}{m}\sum_{j=1}^{m}[\nabla \boldsymbol{L}^{\mu_j(t)}\left(\mathbf{w}_j(t)\right) - \nabla \boldsymbol{L}\left(\mathbf{w}_a(t)\right)]$, and $W(t)$ is a standard Brownian motion (Feynman, 1964) in $\mathbb{R}^d$. We then utilize the SDE approximation of D-SGD to study the escaping efficiency of D-SGD, defined as follows.

**Definition 1** (Escaping efficiency). *Let $\mathbf{w}^*$ denote one of the local minimum of the loss function $\boldsymbol{L}$. Then, we call $\mathbb{E}_{\mathbf{w}_a(t)}[\boldsymbol{L}(\mathbf{w}_a(t)) - \boldsymbol{L}(\mathbf{w}^*)]$ the escaping efficiency of the dynamic $\mathbf{w}_a(t+1)$ from $\mathbf{w}^*$, where $\mathbb{E}_{\mathbf{w}_a(t)}$ denotes the expectation with respect to the distribution of $\mathbf{w}_a(t)$.*

Suppose that $\mathbf{w}_a(t+1)$ gets stuck in a minimum $\mathbf{w}^*$[5], the escaping efficiency characterizes the probability that the dynamics $\mathbf{w}_a(t+1)$ escapes $\mathbf{w}^*$, since Markov's inequality guarantees $\forall \delta$, $P(\boldsymbol{L}(\mathbf{w}_a(t+1)) - \boldsymbol{L}(\mathbf{w}^*) \geq \delta) \leq \big[\mathbb{E}_{\mathbf{w}_a(t)}[\boldsymbol{L}(\mathbf{w}_a(t+1)) - \boldsymbol{L}(\mathbf{w}^*)]\big]/\delta$.

We then have the following theorem on the escaping efficiency of D-SGD.

**Theorem 3** (Escaping efficiency of D-SGD). *If the loss $\boldsymbol{L}$ is continuous and has fourth-order partial derivatives, the escaping efficiency of D-SGD from minimum $\mathbf{w}^*$ satisfies*

$$\mathbb{E}_{\mathbf{w}_a(t)}[\boldsymbol{L}(\mathbf{w}_a(t)) - \boldsymbol{L}(\mathbf{w}^*)]$$
$$= -\int_0^t \mathbb{E}_{\mathbf{w}_a(t)}[\nabla \boldsymbol{L}(\mathbf{w}_a(t))^T \nabla \boldsymbol{L}(\mathbf{w}_a(t)) - \frac{1}{2}grandsum((\boldsymbol{T}(\mathbf{w}_a(t))\nabla \boldsymbol{L}(\mathbf{w}_a(t))) \odot \boldsymbol{\Xi}^*(t))]\mathrm{d}t$$
$$+ \int_0^t \frac{\eta}{2}\operatorname{Tr}\left(\boldsymbol{H}(\mathbf{w}_a(t))\boldsymbol{\Sigma}_{\mathrm{D}(t)}\right)\mathrm{d}t,$$

*where $\odot$ denotes the Hadamard product (Davis, 1962), and grandsum($\cdot$) (Merikoski, 1984) of a matrix $\tilde{\boldsymbol{M}}$ satisfies $grandsum(\tilde{\boldsymbol{M}}) = \sum_{i,j}\tilde{\boldsymbol{M}}_{ij}$.*

A detailed proof and the escaping efficiency of C-SGD (see Proposition C.5) are given in Appendix C.

Comparing Theorem 3 and Proposition C.5, we can see that the main difference between the escaping efficiency of D-SGD and C-SGD lies in the integral of $grandsum((\boldsymbol{T}(\mathbf{w}_a(t))\nabla \boldsymbol{L}(\mathbf{w}_a(t))) \odot \boldsymbol{\Xi}^*(t))$, which correlates with the gradient diversity in Equation (4). We then study how this term affects the escaping efficiency of D-SGD on super-quadratic minima, a typical class of minima as defined below.

**Definition 2** (Super-quadratic minimum). *Given that the loss $\boldsymbol{L}$ is continuous and has second-order partial derivatives, we call the mimimum $\mathbf{w}^*$ of $\boldsymbol{L}$ $\delta$-locally super-quadratic if for any $\mathbf{w}$ in the open punctured neighbourhood of $\mathbf{w}^*$ (i.e., $\mathbf{w} \in \mathring{U}(\mathbf{w}^*, \delta)$), the following condition holds: (1) $\boldsymbol{H}(\mathbf{w}^*) \preccurlyeq \boldsymbol{H}(\mathbf{w})$; and (2) $\exists \alpha(\mathbf{w}), \beta(\mathbf{w}) \in \mathbb{R}^+$ s.t. $\boldsymbol{H}(\mathbf{w})(\mathbf{w} - \mathbf{w}^*) = \alpha(\mathbf{w})(\|\mathbf{w} - \mathbf{w}^*\|_2^{\beta(\mathbf{w})}(\mathbf{w} - \mathbf{w}^*))$.*

---

[5]Note that there is no guarantee that D-SGD can converge to any local minimum in the non-convex settings.

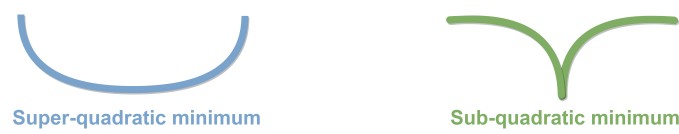

Figure 3: An illustration of super-quadratic and sub-quadratic minimum.

The super-quadratic growth implies that the losses become flatter when the parameters get closer to minima. We then present the intuition of the second condition in Definition 2. A second-order Taylor approximation of $\boldsymbol{L}$ around $\mathbf{w}^*$ reads,

$$\boldsymbol{L}(\mathbf{w}) - \boldsymbol{L}(\mathbf{w}^*) = \nabla\boldsymbol{L}(\mathbf{w})^T(\mathbf{w} - \mathbf{w}^*) + (\mathbf{w} - \mathbf{w}^*)^T\boldsymbol{H}(\mathbf{w})(\mathbf{w} - \mathbf{w}^*),$$

and the second condition in Definition 2 further guarantees that,

$$\boldsymbol{L}(\mathbf{w}) - \boldsymbol{L}(\mathbf{w}^*) = \nabla\boldsymbol{L}(\mathbf{w})^T(\mathbf{w} - \mathbf{w}^*) + \alpha(\mathbf{w})\|\mathbf{w} - \mathbf{w}^*\|_2^{\beta(\mathbf{w})}\underbrace{(\mathbf{w} - \mathbf{w}^*)^T(\mathbf{w} - \mathbf{w}^*)}_{\text{quadratic growth}},$$

which suggests that the growth of $\boldsymbol{L}(\mathbf{w})$ is $\delta$-locally super-quadratic as long as $\alpha(\mathbf{w}), \beta(\mathbf{w}) > 0$.

A related study by Ma et al. (2022) observes that the minima learned by centralized gradient descent methods obey a "sub-quadratic growth" (i.e., the loss becomes sharper as parameters get closer to the minimum). We also give a formalization of the sub-quadratic minima in Definition C.1. Intuitively, super-quadratic minima are flatter than sub-quadratic minima with the same depth, as illustrated in Figure 3. The following proposition studies the sign of grandsum$((\boldsymbol{T}(\mathbf{w}_{a(t)})\nabla\boldsymbol{L}(\mathbf{w}_{a(t)})) \odot \boldsymbol{\Xi}^*(t))$ on the super-quadratic and sub-quadratic minima.

**Proposition 4.** *Suppose that $\mathbf{w}_{a(t)}$ is sufficiently close to a local minimum $\mathbf{w}^*$, grandsum$((\boldsymbol{T}(\mathbf{w}_{a(t)})\nabla\boldsymbol{L}(\mathbf{w}_{a(t)})) \odot \boldsymbol{\Xi}^*(t))$ is (1) zero if $\mathbf{w}^*$ is a quadratic minima, (2) positive if $\mathbf{w}^*$ is a $\delta$-locally super-quadratic minima, and (3) negative if $\mathbf{w}^*$ is a $\delta$-locally sub-quadratic minima.*

Combined with Theorem 3, Proposition 4 shows that D-SGD favors super-quadratic minima over sub-quadratic minima with a higher probability. The proof is included in Appendix C.

Theorem 1 and Proposition 4 indicate that the additional noise (i.e., the gradient diversity in Equation (4)) of D-SGD may play a distinct role in shaping the generalizability of D-SGD.

## 5 EMPIRIAL RESULTS

This section empirically validates our theory. We first introduce the experimental setup and then study how decentralization favours the flatness of minima.

**Implementation settings.** Vanilla D-SGD and C-SGD are employed to train image classifiers on CIFAR-10 (Krizhevsky et al., 2009) with AlexNet (Krizhevsky et al., 2017), ResNet-18 and ResNet-34 (He et al., 2016b), three popular neural networks. Batch normalization (Ioffe & Szegedy, 2015) is employed in training AlexNet. The number of workers (one GPU as a worker) is set as 16; and the local batch size is set as 8, 64, and 512 per worker in three different cases. For the case of local batch size 64, the initial learning rate is set as 0.1 for ResNet-18 and 0.01 for AlexNet. The learning rate is divided by 10 when the model has passed the 2/5 and 4/5 of the total number of iterations (He et al., 2016a). We apply the linear scaling law to avoid different total batch sizes caused by the different local batch size (see Subsection 4.2). In order to understand the effect of decentralization on the flatness of minima, all other training techniques are strictly controlled. The code is written based on PyTorch (Paszke et al., 2019).

**Hardware enviornment.** The experiments are conducted on a computing facility with NVIDIA® Tesla™ V100 16GB GPUs and Intel® Xeon® Gold 6140 CPU @ 2.30GHz CPUs.

We plot the minima learned by C-SGD and D-SGD in Figure 4 using the loss landscape 3D visualization tool in Li et al. (2018). See more plots in Appendix B. Two observations are obtained from these figures: (1) the minima of D-SGD are flatter than those of C-SGD; and (2) the gap in flatness becomes larger as the total batch size increases. These observations support the claims in Theorem 1 and Theorem 2 that D-SGD favors flatter minima than C-SGD, especially in the large-batch settings.

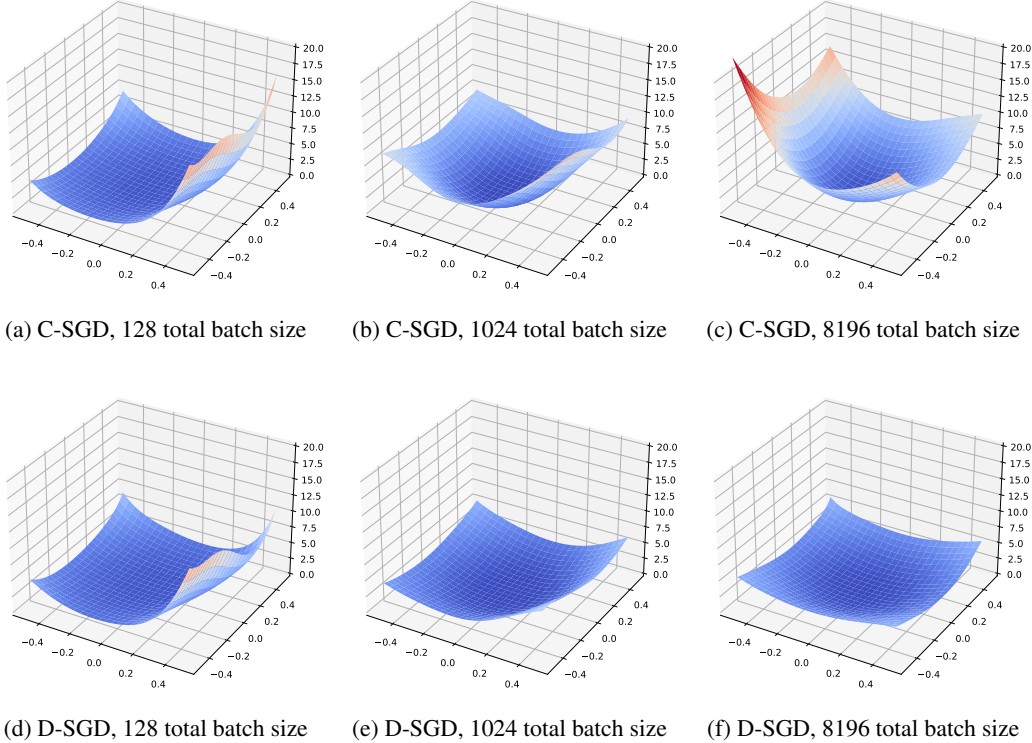

| (a) C-SGD, 128 total batch size | (b) C-SGD, 1024 total batch size | (c) C-SGD, 8196 total batch size |
| (d) D-SGD, 128 total batch size | (e) D-SGD, 1024 total batch size | (f) D-SGD, 8196 total batch size |

Figure 4: Minima 3D visualization of C-SGD and D-SGD with ResNet-18 on CIFAR-10.

## 6 DISCUSSION AND FUTURE WORK

**Scalability to complex or sparse topologies.** Our theory holds for arbitrary topologies (see Definition A.1). We also conduct experiments on grid-like and static exponential topologies (Ying et al., 2021a) and obtain results similar to Figure 4 and Figure B.1. For spare topologies, which has a very small spectral gap, the regularization term in Theorem 1 would be extremely large during training, which may hinder optimization and lead to a large total excess risk of D-SGD. Can we design a new decentralized training algorithm that can alleviate the optimization issue on spare topologies while maintaining the generalization advantage in large-batch setting?

**Non-IIDness and the flatness of minima.** In real-world settings, a fundamental challenge in distributed learning is that data may not be i.i.d. across workers (Tang et al., 2018; Vogels et al., 2021; Mendieta et al., 2022). In this case, different workers may collect distinct or even contradictory samples (i.e., data-heterogeneity) (Criado et al., 2021). It is widely observed that the non-IIDness hurts the generalizability of D-SGD. Can we rigorously analyze how the degree of data-heterogeneity affects the flatness of minima and design theoretically motivated algorithms to promote the generalizability of D-SGD in non-IID settings?

## 7 CONCLUSION

This work provides a new theory that reconciles the conflict between the empirical observations showing that D-SGD can generalize better than centralized SGD (C-SGD) in large-batch settings and the existing generalization theories of D-SGD which suggest that decentralization degrades generalizability. We prove that D-SGD introduces an implicit regularization that penalizes the learned minima's sharpness and this effect will be amplified in large-batch settings if we apply the linear scaling rule. We further analyze the escaping efficiency of D-SGD, which shows that D-SGD favors super-quadratic flat minima. To our best knowledge, this is the first work on the implicit sharpness regularization and escaping efficiency of D-SGD.

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

# A    ADDITIONAL BACKGROUND

## A.1    DECENTRALIZED LEARNING

To handle an increasing amount of data and model parameters, distributed learning across multiple computing workers emerges. A traditional distributed learning system usually follows a centralized setup. However, such a central server-based learning scheme suffers from two main issues: (1) A centralized communication protocol significantly slows down training since central servers are easily overloaded, especially in low-bandwidth or high-latency cases (Lian et al., 2017); (2) There exists a potential information leakage through privacy attacks on model parameters despite decentralizing data using Federated Learning (Zhu et al., 2019a; Geiping et al., 2020; Yin et al., 2021). As an alternative, decentralized training allows workers to balance the load on the central server through the gossip technique (Lian et al., 2017), as well as maintain confidentiality (Warnat-Herresthal et al., 2021).

We then summarize some commonly used notions regarding decentralized learning.

**Definition A.1** (Doubly Stochastic Matrix). *Let $\mathcal{G} = (\mathcal{V}, \mathcal{E})$ stand for the decentralized communication topology where $\mathcal{V}$ denotes the set of $m$ computational nodes and $\mathcal{E}$ represents the edge set. For any given topology $\mathcal{G} = (\mathcal{V}, \mathcal{E})$, the doubly stochastic gossip matrix $\mathbf{P} = [\mathbf{P}_{j,k}] \in \mathbb{R}^{m \times m}$ is defined on the edge set $\mathcal{E}$ that satisfies*

- $\mathbf{P} = \mathbf{P}^T$ *(symmetric)*;

- *If $j \neq k$ and $(j, k) \notin \mathcal{E}$, then $\mathbf{P}_{j,k} = 0$ (disconnected) and otherwise, $\mathbf{P}_{j,k} > 0$ (connected);*

- $\mathbf{P}_{j,k} \in [0, 1]$ $\forall k, l$ *and $\sum_k \mathbf{P}_{j,k} = \sum_l \mathbf{P}_{j,k} = 1$ (standard weight matrix for undirected graph).*

In the following we illustrate some commonly-used communication topologies.

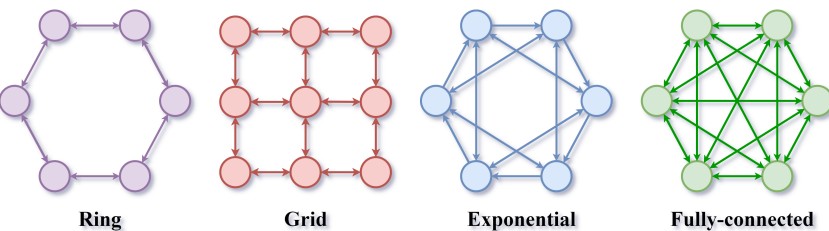

Figure A.1: An illustration of some commonly used topologies.

The intensity of gossip communications is measured by the spectral gap (Seneta, 2006) of $\mathbf{P}$.

**Definition A.2** (Spectral Gap). *Denote $\lambda = \max\{|\lambda_2|, |\lambda_m|\}$ where $\lambda_i$ ($i = 2, \ldots, m$) is the $i$-th largest eigenvalue of gossip matrix $\mathbf{P} \in \mathbb{R}^{m \times m}$. The spectral gap of a gossip matrix $\mathbf{P}$ can be defined as follows:*

$$spectral\ gap := 1 - \lambda.$$

*According to the definition of doubly stochastic matrix (Definition A.1), we have $0 \leq \lambda < 1$. The spectral gap measures the connectivity of the communication topology, which is close to 0 for sparse topologies and will approach 1 for well-connected topologies.*

**Assumption A.1.** *We assume that the sum of the off-diagonal entries of $\Xi(t)$ is smaller than $d - 1$ times of the sum of the diagonal entries of $\Xi(t)$ in expectation:*

$$\mathbb{E}_{\substack{\mu_j(\tau) \sim D \\ j=1,\ldots,m \\ \tau=1,\cdots,t-1}} \left( \sum_{k \neq l} \frac{1}{m} \sum_{j=1}^{m} (\mathbf{w}_j(t) - \mathbf{w}_a(t))_k (\mathbf{w}_j(t) - \mathbf{w}_a(t))_l \right) \leq \mathbb{E}_{\substack{\mu_j(\tau) \sim D \\ j=1,\ldots,m \\ \tau=1,\cdots,t}} \left( (d-1) \sum_{k=1}^{m} \frac{1}{m} \sum_{j=1}^{m} (\mathbf{w}_j(t) - \mathbf{w}_a(t))_k^2 \right),$$

*where $d$ stands for the dimensionality of $\mathbf{w}_j(t) - \mathbf{w}_a(t)$.*

## A.2 EXPLANATION OF TENSOR PRODUCT

The tensor product between a third-order tensor $\boldsymbol{T} \in \mathbb{R}^{d \times d \times d}$ and a second-order tensor (matrix) $\boldsymbol{M} \in \mathbb{R}^{d \times d}$ in this paper is defined as

$$\underbrace{(\boldsymbol{T} \otimes \boldsymbol{M})_i}_{\text{the } i\text{-th entry}} = \text{grandsum}(\boldsymbol{T}_i \odot \boldsymbol{M}),$$

where $\boldsymbol{T}_i \in \mathbb{R}^{d \times d}$ is a second-order tensor (matrix), $\odot$ denotes the Hadamard product (Davis, 1962), and the grandsum($\cdot$) (Merikoski, 1984) of a second-order tensor (matrix) $\tilde{\boldsymbol{M}}$ satisfies grandsum($\tilde{\boldsymbol{M}}$) $= \sum_{i,j} \tilde{\boldsymbol{M}}_{ij}$.

# B ADDITIONAL MINIMA VISUALIZATION

We plot the minima learned by C-SGD and D-SGD as follows using the 2D loss landscape visualization tool in Li et al. (2018).

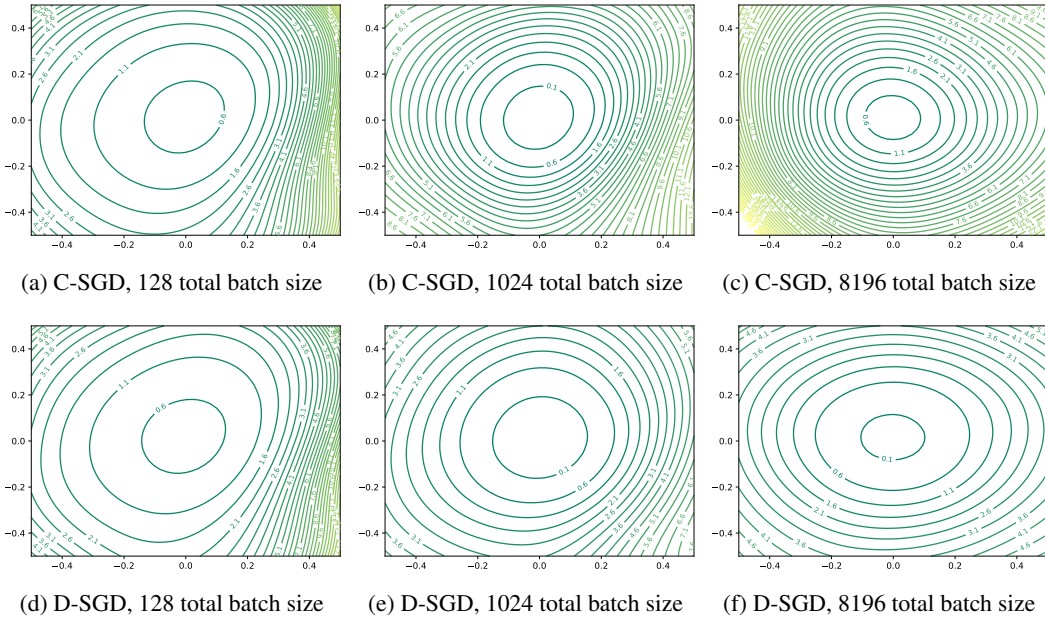

(a) C-SGD, 128 total batch size    (b) C-SGD, 1024 total batch size    (c) C-SGD, 8196 total batch size

(d) D-SGD, 128 total batch size    (e) D-SGD, 1024 total batch size    (f) D-SGD, 8196 total batch size

Figure B.1: Minima 2D visualization of C-SGD and D-SGD with ResNet-18 on CIFAR-10.

From Figure B.1, we observe that (1) the minima of D-SGD is flatter than those of C-SGD; and (2) the gap in flatness becomes larger as the total batch size increases. Similar results are obtained with AlexNet and ResNet-34 on CIFAR-10.

## C  PROOF

**Corollary C.1.** *The gradient diversity in [Equation (4)] equals to zero in the following cases: (1) the loss $\boldsymbol{L} = \mathbf{w}^T \boldsymbol{H} \mathbf{w}$ is quadratic; and (2) the optimization algorithm is distributed centralized SGD ([Equation (1)]),*

*Proof of [Corollary C.1].*

On quadratic loss, we have

$$\frac{1}{m} \sum_{j=1}^{m} [\nabla \boldsymbol{L}^{\mu_j(t)}(\mathbf{w}_j(t)) - \nabla \boldsymbol{L}^{\mu_j(t)}(\mathbf{w}_a(t))] = \frac{1}{m} \sum_{j=1}^{m} [\boldsymbol{H}\mathbf{w}_j(t) - \boldsymbol{H}\mathbf{w}_a(t)] = \boldsymbol{H} \underbrace{\frac{1}{m} \sum_{j=1}^{m} [\mathbf{w}_j(t) - \mathbf{w}_a(t)]}_{0}.$$

In distributed centralized SGD, the gradient diversity statisfies

$$\frac{1}{m} \sum_{j=1}^{m} [\nabla \boldsymbol{L}^{\mu_j(t)}(\underbrace{\mathbf{w}_j(t)}_{=\mathbf{w}_a(t)}) - \nabla \boldsymbol{L}^{\mu_j(t)}(\mathbf{w}_a(t))] = 0.$$

$\square$

**Lemma C.2.** *We denote $\boldsymbol{\Xi}(t) = \frac{1}{m} \sum_{j=1}^{m} (\mathbf{w}_j(t) - \mathbf{w}_a(t))(\mathbf{w}_j(t) - \mathbf{w}_a(t))^T$ the weight diversity matrix and $\boldsymbol{\Xi}^*(t) = \sum_{i=1}^{d} \langle \mathrm{e}_i \mathrm{e}_i^T, \boldsymbol{\Xi}(t) \rangle_F \; \mathrm{e}_i \mathrm{e}_i^T$. We assume that $d_1$ $(d_1 < d)$, the marginal contribution of $\lambda_{H(\mathbf{w}_a(t)),1}$ on the full spectrum of $H(\mathbf{w}_a(t))$, is non-negative and satisfies $\lambda_{H(\mathbf{w}_a(t)),1} = \frac{d_1}{d} \mathrm{Tr}(H(\mathbf{w}_a(t)))$. Then the product of $\mathrm{Tr}(\boldsymbol{\Xi}(t))$ and the maximum eigenvalue of $H(\mathbf{w}_a(t))$ is upper and lower bounded as*

$$0 \le \mathrm{Tr}(H(\mathbf{w}_a(t))\boldsymbol{\Xi}^*(t)) \le \underbrace{\lambda_{\boldsymbol{H}(\mathbf{w}_a(t)),1}}_{sharpness} \cdot \underbrace{\mathrm{Tr}(\boldsymbol{\Xi}(t))}_{consensus\ distance} \le d_1 \mathrm{Tr}(H(\mathbf{w}_a(t))\boldsymbol{\Xi}^*(t)).$$

*Proof of [Lemma C.2].*

On the one hand, von Neumann's trace inequality ([Von Neumann], [1937]) guarantees

$$\mathrm{Tr}(\boldsymbol{H}(\mathbf{w}_a(t))\boldsymbol{\Xi}^*(t)) \le \sum_{r=1}^{d} \lambda_{\boldsymbol{H}(\mathbf{w}_a(t)),r} \cdot \lambda_{\boldsymbol{\Xi}^*(t),r} \le \lambda_{\boldsymbol{H}(\mathbf{w}_a(t)),1} \cdot \mathrm{Tr}(\boldsymbol{\Xi}(t)), \qquad \text{(C.1)}$$

where $\lambda_{\boldsymbol{H}(\mathbf{w}_a(t)),r}$ and $\lambda_{\boldsymbol{\Xi}(t),r}$ represent the $r$-th largest eigenvalue of $\boldsymbol{H}(\mathbf{w}_a(t))$ and $\boldsymbol{\Xi}(t)$, respectively. On the other hand, we will prove that $\lambda_{\boldsymbol{H}(\mathbf{w}_a(t)),1} \cdot \mathrm{Tr}(\boldsymbol{\Xi}(t)) \le \mathcal{O}(\mathrm{Tr}(\boldsymbol{H}(\mathbf{w}_a(t))\boldsymbol{\Xi}^*(t)))$. Since $\boldsymbol{\Xi}^*(t) = \sum_{i=1}^{d} \langle \mathrm{e}_i \mathrm{e}_i^T, \boldsymbol{\Xi}(t) \rangle_F \; \mathrm{e}_i \mathrm{e}_i^T$ is a diagonal matrix, $\mathrm{Tr}(\boldsymbol{H}(\mathbf{w}_a(t))\boldsymbol{\Xi}^*(t))$ can be lower bounded as

$$\mathrm{Tr}(\boldsymbol{H}(\mathbf{w}_a(t))\boldsymbol{\Xi}^*(t)) \ge \xi^2 \mathrm{Tr}(\boldsymbol{H}(\mathbf{w}_a(t))),$$

where $\xi^2$ is the lower bound of $\frac{1}{m} \sum_{j=1}^{m} (\mathbf{w}_j(t) - \mathbf{w}_a(t))_k^2$ $(k = 1, \ldots, d)$.

Knowing that $\mathrm{Tr}(\boldsymbol{\Xi}^*(t)) = \mathrm{Tr}(\boldsymbol{\Xi}(t))$, we can bound the right hand side of [Equation (C.1)] as follows:

$$0 \le \lambda_{\boldsymbol{H}(\mathbf{w}_a(t)),1} \cdot \mathrm{Tr}(\boldsymbol{\Xi}(t)) \le \frac{d_1}{d\xi^2} \mathrm{Tr}(\boldsymbol{H}(\mathbf{w}_a(t))\boldsymbol{\Xi}^*(t)) \cdot \mathrm{Tr}(\boldsymbol{\Xi}(t)) \le d_1 \mathrm{Tr}(\boldsymbol{H}(\mathbf{w}_a(t))\boldsymbol{\Xi}^*(t)).$$

Note than we can also obtain

$$0 \le \mathrm{Tr}(\boldsymbol{H}(\mathbf{w}_a(t))\boldsymbol{\Xi}^*(t)) \le \mathrm{Tr}(\boldsymbol{H}(\mathbf{w}_a(t))) \cdot \mathrm{Tr}(\boldsymbol{\Xi}(t)) \le d_1 \mathrm{Tr}(\boldsymbol{H}(\mathbf{w}_a(t))\boldsymbol{\Xi}^*(t)),$$

which shows that D-SGD also implicitly regularizes $\mathrm{Tr}(\boldsymbol{H}(\mathbf{w}_a(t)))$.

$\square$

**Lemma C.3** (([Kong et al.], [2021]))**.** *Suppose that the averaged gradient norm satisfies $\frac{1}{m} \sum_{j=1}^{m} \|\nabla \boldsymbol{L}(\mathbf{w}_j(t))\|^2 \le (1 + \frac{1-\lambda}{4}) \frac{1}{m} \sum_{j=1}^{m} \|\nabla \boldsymbol{L}(\mathbf{w}_j(t+1))\|^2$, then the the consensus distance of D-SGD satisfies*

$$\mathrm{Tr}(\boldsymbol{\Xi}(t)) = \frac{1}{m} \sum_{j=1}^{m} \|\mathbf{w}_j(t) - \mathbf{w}_a(t)\|_2^2$$

$$= \lambda \eta^2 \cdot \mathcal{O}\left( \frac{\frac{1}{m}\sum_{j=1}^m \|\nabla \boldsymbol{L}\left(\mathbf{w}_j(t)\right)\|^2}{(1-\lambda)^2} + \frac{\frac{1}{m}\sum_{j=1}^m \mathbb{E}_{\mu_j(t)\sim\mathcal{D}}\left\|\nabla \boldsymbol{L}^{\mu_j(t)}\left(\mathbf{w}_j(t)\right) - \nabla \boldsymbol{L}\left(\mathbf{w}_j(t)\right)\right\|_2^2}{1-\lambda} \right),$$

*where $\lambda$ equals to $1 - spectral\ gap$ (see Definition A.2).*

**Lemma C.4.** *D-SGD is approximated by the following SDE*

$$\mathrm{d}\mathbf{w}_a(t) = -\left[\nabla \boldsymbol{L}\left(\mathbf{w}_a(t)\right) + \frac{1}{2}\boldsymbol{T}(\mathbf{w}_a(t)) \otimes \boldsymbol{\Xi}^*(t)\right]\mathrm{d}t + \sqrt{\eta \boldsymbol{\Sigma}_D(t)}\mathrm{d}W(t),$$

*where $\otimes$ denotes the tensor product (see Appendix A.2), $\boldsymbol{\Sigma}_D(t)$ denotes the covariance matrix of the unbiased noise $\epsilon(t)$, and $W(t)$ is a standard Brownian motion (Feynman, 1964) in $\mathbb{R}^d$.*

*Proof of Lemma C.4.*

If we omit the residual terms, the iterate of D-SGD becomes

$$\mathbf{w}_a(t+1) = \mathbf{w}_a(t) - \eta\nabla\left[\boldsymbol{L}\left(\mathbf{w}_a(t)\right) + \mathrm{Tr}(\boldsymbol{H}(\mathbf{w}_a(t))\boldsymbol{\Xi}^*(t))\right] + \eta\epsilon_D(t)$$

$$= \mathbf{w}_a(t) - \left[\nabla \boldsymbol{L}\left(\mathbf{w}_a(t)\right) + \frac{1}{2}\boldsymbol{T}(\mathbf{w}_a(t)) \otimes \boldsymbol{\Xi}^*(t)\right]\eta + \sqrt{\eta \boldsymbol{\Sigma}_D(t)}\sqrt{\eta}\epsilon^*,$$

where $\epsilon_D(t) \sim \mathcal{N}\left(0, \boldsymbol{\Sigma}_D(t)\right)$ (Gaussian approximation) and $\epsilon^*$ is a standard Gaussian random variable.

For small enough constant learning rate $\eta$, we arrive at

$$\mathrm{d}\mathbf{w}_a(t) = -\left[\nabla \boldsymbol{L}\left(\mathbf{w}_a(t)\right) + \frac{1}{2}\boldsymbol{T}(\mathbf{w}_a(t)) \otimes \boldsymbol{\Xi}^*(t)\right]\mathrm{d}t + \sqrt{\eta \boldsymbol{\Sigma}_D(t)}\mathrm{d}W(t).$$

The stochastic processes give a way to model D-SGD as a continuous-time evolution (i.e., SDE) without ignoring the role of mini-batch noise if the learning rate is infinitesimal.

□

**Theorem 1** (Implicit regularization of D-SGD). *Given the loss $\boldsymbol{L}$ is continuous and has fourth-order partial derivatives. Denote the weight diversity matrix as $\boldsymbol{\Xi}(t) = \frac{1}{m}\sum_{j=1}^m \left(\mathbf{w}_j(t) - \mathbf{w}_a(t)\right)\left(\mathbf{w}_j(t) - \mathbf{w}_a(t)\right)^T$, its diagonal matrix as $\boldsymbol{\Xi}^*(t)$, and the d-dimensional all-ones vector as $\mathbf{1}$. With probability greater than $1 - \mathcal{O}(\eta)$, the mean iterate of D-SGD becomes*

$$\mathbb{E}_{\substack{\mu_j(t)\sim D \\ j=1,\ldots,m}}\left[\mathbf{w}_a(t+1)\right]$$

$$= \mathbf{w}_a(t) - \eta\nabla\underbrace{\left[\boldsymbol{L}\left(\mathbf{w}_a(t)\right) + \frac{1}{2}\mathrm{Tr}(\boldsymbol{H}(\mathbf{w}_a(t))\boldsymbol{\Xi}^*(t))\right]}_{\text{the regularized loss}} + \mathcal{O}(\eta^{\frac{1}{2}}\mathbf{1}) + \mathcal{O}\left(\eta\|\mathbf{w}_j(t) - \mathbf{w}_a(t)\|_2^3 \mathbf{1}\right),$$

*Under the mild assumptions in Lemma C.2, D-SGD implicitly regularizes*

$$reg(\underset{j=1,\ldots,m}{\mathbf{w}_j(t)}) = \underbrace{\lambda_{\boldsymbol{H}(\mathbf{w}_a(t)),1}}_{\text{maximum Hessian eigenvalue}} \cdot \underbrace{\mathrm{Tr}(\boldsymbol{\Xi}(t))}_{\text{consensus distance}}.$$

*Proof of Theorem 1.*

We start by rewriting the update of the global averaged model $\mathbf{w}_a(t)$ of D-SGD as follows,

$$\mathbf{w}_a(t+1) = \mathbf{w}_a(t) - \eta\Big[\underbrace{\nabla \boldsymbol{L}\left(\mathbf{w}_a(t)\right)}_{\text{unbiased gradient}} + \underbrace{\nabla \boldsymbol{L}\left(\mathbf{w}_a(t)\right) - \nabla \boldsymbol{L}^{\mu(t)}\left(\mathbf{w}_a(t)\right)}_{\text{gradient noise over the superbatch }\mu(t)}$$

$$+ \underbrace{\frac{1}{m}\sum_{j=1}^m [\nabla \boldsymbol{L}^{\mu_j(t)}\left(\mathbf{w}_j(t)\right) - \nabla \boldsymbol{L}^{\mu_j(t)}\left(\mathbf{w}_a(t)\right)]}_{\text{gradient diversity among workers}}\Big].$$

Analyzing the effect of the gradient diversity on the training dynamics of D-SGD on the general non-convex losses is highly non-trivial. Technically, we perform a second-order Taylor expansion (see Appendix A.2) on the gradient diversity around $\mathbf{w}_a(t)$, omitting the high-order residuals $R$:

$$\frac{1}{m}\sum_{j=1}^m [\nabla \boldsymbol{L}^{\mu_j(t)}\left(\mathbf{w}_j(t)\right) - \nabla \boldsymbol{L}^{\mu_j(t)}\left(\mathbf{w}_a(t)\right)]$$

$$= \frac{1}{m} \sum_{j=1}^{m} \boldsymbol{H}^{\mu_j(t)}(\mathbf{w}_a(t))(\mathbf{w}_j(t) - \mathbf{w}_a(t)) + \frac{1}{2m} \sum_{j=1}^{m} \boldsymbol{T}^{\mu_j(t)}(\mathbf{w}_a(t)) \otimes [(\mathbf{w}_j(t) - \mathbf{w}_a(t))(\mathbf{w}_j(t) - \mathbf{w}_a(t))^T].$$

Here $\boldsymbol{H}^{\mu_j(t)}(\mathbf{w}_a(t)) \triangleq \frac{1}{|\mu_j(t)|} \sum_{\zeta(t)=1}^{|\mu_j(t)|} \boldsymbol{H}(\mathbf{w}_a(t); z_{j,\zeta(t)})$ stands for the empirical Hessian at $\mathbf{w}_a(t)$ and $\boldsymbol{T}^{\mu_j(t)}(\mathbf{w}_a(t)) \triangleq \frac{1}{|\mu_j(t)|} \sum_{\zeta(t)=1}^{|\mu_j(t)|} \boldsymbol{T}(\mathbf{w}_a(t); z_{j,\zeta(t)})$ denotes the tensor containing all empirical third-order partial derivatives at $\mathbf{w}_a(t)$, where $\mu_j(t)$ and $z_{j,\zeta(t)}$ follows the notation in Equation (1).

Analogous to the works investigating the SGD dynamics (M et al., 2017; Zhu et al., 2019b; Ziyin et al., 2022; Wu et al., 2022), we will calculate the expectation and covariance of the gradient diversity. The expectation of gradient diversity is first calculated as follows. We defer the analysis of its covariance to Subsection 4.3. Taking expectation over all local mini-batches $\mu_{j(t)}$ $(j = 1, \ldots, m)$ provides

$$\mathbb{E}_{\substack{\mu_j(t) \sim D \\ j=1,\ldots,m}} \Big[ \frac{1}{m} \sum_{j=1}^{m} [\nabla \boldsymbol{L}^{\mu_j(t)}(\mathbf{w}_j(t)) - \nabla \boldsymbol{L}^{\mu_j(t)}(\mathbf{w}_a(t))] \Big]$$

$$= \boldsymbol{H}(\mathbf{w}_a(t)) \underbrace{\frac{1}{m} \sum_{j=1}^{m} (\mathbf{w}_j(t) - \mathbf{w}_a(t))}_{=0} + \frac{1}{2} \boldsymbol{T}(\mathbf{w}_a(t)) \otimes \Big[ \frac{1}{m} \sum_{j=1}^{m} (\mathbf{w}_j(t) - \mathbf{w}_a(t))(\mathbf{w}_j(t) - \mathbf{w}_a(t))^T \Big] + R.$$

The $i$-th entry of the above equation will be

$$\mathbb{E}_{\substack{\mu_j(t) \sim D \\ j=1,\ldots,m}} \Big[ \frac{1}{m} \sum_{j=1}^{m} [\partial_i \boldsymbol{L}^{\mu_j(t)}(\mathbf{w}_j(t)) - \partial_i \boldsymbol{L}^{\mu_j(t)}(\mathbf{w}_a(t))] \Big]$$

$$= \frac{1}{2} \underbrace{\sum_{k,l} \partial_{ikl}^3 \boldsymbol{L}(\mathbf{w}_a(t)) \frac{1}{m} \sum_{j=1}^{m} (\mathbf{w}_j(t) - \mathbf{w}_a(t))_k (\mathbf{w}_j(t) - \mathbf{w}_a(t))_l}_{= \partial_i \sum_{kl} \partial_{kl}^2 \boldsymbol{L}(z_n) \frac{1}{m} \sum_{j=1}^{m} (\mathbf{w}_j(t) - \mathbf{w}_a(t))_k (\mathbf{w}_j(t) - \mathbf{w}_a(t))_l} + \mathcal{O}\left( \|\mathbf{w}_j(t) - \mathbf{w}_a(t)\|_2^3 \right), \quad \text{(C.2)}$$

where $(\mathbf{w}_j(t) - \mathbf{w}_a(t))_k$ denotes the $k$-th entry of the vector $\mathbf{w}_j(t) - \mathbf{w}_a(t)$. The equality in the brace is due to Clairaut's theorem (Rudin et al., 1976).

According to Markov's inequality and Assumption A.1, we obtain

$$\Pr\Big\{ \sum_{k \neq l} \frac{1}{m} \sum_{j=1}^{m} (\mathbf{w}_j(t) - \mathbf{w}_a(t))_k (\mathbf{w}_j(t) - \mathbf{w}_a(t))_l > \eta^{\frac{1}{2}} \Big\}$$

$$\leq \frac{\mathbb{E}_{\substack{\mu_j(\tau) \sim D \\ j=1,\ldots,m \\ \tau=1,\cdots,t-1}} \left( \sum_{k \neq l} \frac{1}{m} \sum_{j=1}^{m} (\mathbf{w}_j(t) - \mathbf{w}_a(t))_k (\mathbf{w}_j(t) - \mathbf{w}_a(t))_l \right)}{\eta^{\frac{1}{2}}}$$

$$\leq \frac{\mathbb{E}_{\substack{\mu_j(\tau) \sim D \\ j=1,\ldots,m \\ \tau=1,\cdots,t-1}} \left( (d-1) \sum_{k=1}^{m} \frac{1}{m} \sum_{j=1}^{m} (\mathbf{w}_j(t) - \mathbf{w}_a(t))_k^2 \right)}{\eta^{\frac{1}{2}}}$$

$$\leq \frac{\mathbb{E}_{\substack{\mu_j(\tau) \sim D \\ j=1,\ldots,m \\ \tau=1,\cdots,t-1}} \left( (d-1) \operatorname{Tr}(\boldsymbol{\Xi}(t)) \right)}{\eta}$$

$$= \frac{\mathcal{O}(d\eta^2)}{\eta^{\frac{1}{2}}}$$

$$= \mathcal{O}(d\eta^{\frac{3}{2}}),$$

where $d$ stands for the dimensionality of $\mathbf{w}_j(t) - \mathbf{w}_a(t)$ and the penultimate equality is due to Lemma C.3.

For sufficiently small $\eta = o(d^{-2})$, $\frac{1}{2} \partial_i \sum_{kl} \partial_{kl}^2 \boldsymbol{L}(z_n) \frac{1}{m} \sum_{j=1}^{m} (\mathbf{w}_j(t) - \mathbf{w}_a(t))_k (\mathbf{w}_j(t) - \mathbf{w}_a(t))_l$ in Equation (C.2) is of the order $\mathcal{O}(\eta)$.

Then we derive that with probability greater than $1-\mathcal{O}(\eta)$, the iterate of D-SGD can be written as

$$\mathbb{E}_{\substack{\mu_j(t)\sim D \\ j=1,\ldots,m}}\big[\mathbf{w}_a(t+1)\big]$$

$$=\mathbf{w}_a(t)-\eta\nabla\underbrace{\Big[\boldsymbol{L}\left(\mathbf{w}_a(t)\right)+\frac{1}{2}\operatorname{Tr}(\boldsymbol{H}(\mathbf{w}_a(t))\boldsymbol{\Xi}^*(t))\Big]}_{\text{the regularized loss}}+\mathcal{O}(\eta^2\mathbf{1})+\mathcal{O}\left(\eta\|\mathbf{w}_j(t)-\mathbf{w}_a(t)\|_2^3\mathbf{1}\right),$$

where $\boldsymbol{\Xi}^*(t)=\sum_{i=1}^d\langle \mathrm{e}_i\mathrm{e}_i^T,\boldsymbol{\Xi}(t)\rangle_F\,\mathrm{e}_i\mathrm{e}_i^T$ is the diagonal of $\Xi(t)$.

According to Lemma C.2, $\lambda_{\boldsymbol{H}(\mathbf{w}_a(t)),1}\cdot\operatorname{Tr}(\boldsymbol{\Xi}(t))$ scales positively with $\operatorname{Tr}(\boldsymbol{H}(\mathbf{w}_a(t))\boldsymbol{\Xi}^*(t))$:

$$0\leq\operatorname{Tr}(\boldsymbol{H}(\mathbf{w}_a(t))\boldsymbol{\Xi}^*(t))\leq\underbrace{\lambda_{\boldsymbol{H}(\mathbf{w}_a(t)),1}}_{\text{sharpness}}\cdot\underbrace{\operatorname{Tr}(\boldsymbol{\Xi}(t))}_{\text{consensus distance}}\leq d_1\operatorname{Tr}(\boldsymbol{H}(\mathbf{w}_a(t))\boldsymbol{\Xi}^*(t)),$$

where $\lambda_{\boldsymbol{H}(\mathbf{w}_a(t)),1}$ denotes the largest eigenvalue of $\boldsymbol{H}(\mathbf{w}_a(t))$ and $d_1$ stands for the marginal contribution of $\lambda_{\boldsymbol{H}(\mathbf{w}_a(t)),1}$ on the full spectrum of $\boldsymbol{H}(\mathbf{w}_a(t))$ (i.e., $\lambda_{\boldsymbol{H}(\mathbf{w}_a(t)),1}=\frac{d_1}{d}\operatorname{Tr}(\boldsymbol{H}(\mathbf{w}_a(t)))$). Therefore, combined with Equation (3), we conclude that D-SGD also implicitly regularizes $\lambda_{\boldsymbol{H}(\mathbf{w}_a(t)),1}\cdot\operatorname{Tr}(\boldsymbol{\Xi}(t))$. The proof is complete.

$\square$

**Theorem 2.** *Suppose that the averaged gradient norm satisfies $\frac{1}{m}\sum_{j=1}^m\|\nabla\boldsymbol{L}\left(\mathbf{w}_j(t)\right)\|^2\leq(1+\frac{1-\lambda}{4})\frac{1}{m}\sum_{j=1}^m\|\nabla\boldsymbol{L}\left(\mathbf{w}_j(t+1)\right)\|^2$, where $1-\lambda$ denotes the spectral gap (see Definition A.2). The sharpness regularization coefficient of D-SGD at $t$-th iteration is $\mathcal{O}(|\mu(t)|^2(1+\frac{1}{m}\sum_{j=1}^m\frac{1}{|\mu_j(t)|}))$, which increases with the total batch size $|\mu(t)|$ if we apply the linear scaling rule.*

*Proof of Theorem 2.*

Theorem 1 states that the regularization coefficient of $\lambda_{\boldsymbol{H}(\mathbf{w}_a(t)),1}$ is $\eta\operatorname{Tr}(\boldsymbol{\Xi}(t))$. According to Lemma C.3, $\operatorname{Tr}(\boldsymbol{\Xi}(t))$ satisfies

$$\operatorname{Tr}(\boldsymbol{\Xi}(t))=\eta^2\cdot\mathcal{O}\left(\underbrace{\frac{1}{m}\sum_{j=1}^m\|\nabla\boldsymbol{L}\left(\mathbf{w}_j(t)\right)\|^2}_{\text{independent of total batch size }|\mu(t)|}+\frac{1}{m}\sum_{j=1}^m\underbrace{\mathbb{E}_{\mu_j(t)\sim\mathcal{D}}\left\|\nabla\boldsymbol{L}^{\mu_j(t)}\left(\mathbf{w}_j(t)\right)-\nabla\boldsymbol{L}\left(\mathbf{w}_j(t)\right)\right\|_2^2}_{\text{noise covariance, }\mathcal{O}(\frac{1}{|\mu_j(t)|})\text{ (M et al., 2017)}}\right).$$

$$\text{(C.3)}$$

Given that we apply the linear scaling rule (see Subsection 4.2), we have $\eta=\mathcal{O}(|\mu(t)|)$, which completes the proof.

$\square$

**Theorem 3** (Escaping efficiency of D-SGD). *If the loss $\boldsymbol{L}$ is continuous and has fourth-order partial derivatives, the escaping efficiency of D-SGD from minimum $\mathbf{w}^*$ satisfies*

$$\mathbb{E}_{\mathbf{w}_a(t)}[\boldsymbol{L}(\mathbf{w}_a(t))-\boldsymbol{L}(\mathbf{w}^*)]$$

$$=-\int_0^t\mathbb{E}_{\mathbf{w}_a(t)}[\nabla\boldsymbol{L}(\mathbf{w}_a(t))^T\nabla\boldsymbol{L}(\mathbf{w}_a(t))-\frac{1}{2}grandsum((\boldsymbol{T}(\mathbf{w}_a(t))\nabla\boldsymbol{L}(\mathbf{w}_a(t)))\odot\boldsymbol{\Xi}^*(t))]\mathrm{d}t$$

$$+\int_0^t\frac{\eta}{2}\operatorname{Tr}\left(\boldsymbol{H}(\mathbf{w}_a(t))\boldsymbol{\Sigma}_D(t)\right)\mathrm{d}t,$$

*where $\odot$ denotes the Hadamard product (Davis, 1962), and the grandsum($\cdot$) (Merikoski, 1984) of a matrix $\tilde{\boldsymbol{M}}$ satisfies grandsum($\tilde{\boldsymbol{M}}$) $=\sum_{i,j}\tilde{\boldsymbol{M}}_{ij}$.*

*Proof of Theorem 3.*

Since $\boldsymbol{L}$ is continuous and has second-order partial derivatives, we can write

$$\mathrm{d}\boldsymbol{L}(\mathbf{w}_a(t))=-\left(\nabla\boldsymbol{L}(\mathbf{w}_a(t))^T\nabla\boldsymbol{L}(\mathbf{w}_a(t))-\frac{1}{2}\underbrace{\nabla\boldsymbol{L}(\mathbf{w}_a(t))^T(\boldsymbol{T}(\mathbf{w}_a(t))\otimes\boldsymbol{\Xi}^*(t))}_{\text{grandsum}((\boldsymbol{T}(\mathbf{w}_a(t))\nabla\boldsymbol{L}(\mathbf{w}_a(t)))\odot\boldsymbol{\Xi}^*(t))}\right)\mathrm{d}t$$

$$+ \frac{\eta}{2} \operatorname{Tr}\left(\boldsymbol{\Sigma}_{\mathrm{D}(t)}^{\frac{1}{2}} \boldsymbol{H}(\mathbf{w}_a(t))\boldsymbol{\Sigma}_{\mathrm{D}(t)}^{\frac{1}{2}}\right) \mathrm{d}t + \nabla \boldsymbol{L}(\mathbf{w}_a(t))^T \boldsymbol{\Sigma}_{\mathrm{D}(t)}\mathrm{d}W(t),$$

according to the Ito's lemma (Øksendal, 2003). The term $\nabla \boldsymbol{L}(\mathbf{w}_a(t))^T \boldsymbol{\Sigma}_{\mathrm{D}(t)}\mathrm{d}W(t)$ will be averaged if we take the expectation with respect to the distribution of $\mathbf{w}_a(t)$. Finally, integrating over $t$ will provide

$$\mathbb{E}_{\mathbf{w}_a(t)}[\boldsymbol{L}(\mathbf{w}_a(t))]$$
$$= \boldsymbol{L}(\mathbf{w}^*) + \int_0^t \frac{\eta}{2} \operatorname{Tr}\left(\boldsymbol{H}(\mathbf{w}_a(t))\boldsymbol{\Sigma}_{\mathrm{D}(t)}\right) \mathrm{d}t$$
$$- \int_0^t \mathbb{E}_{\mathbf{w}_a(t)}[\nabla \boldsymbol{L}(\mathbf{w}_a(t))^T \nabla \boldsymbol{L}(\mathbf{w}_a(t)) - \frac{1}{2}\mathrm{grandsum}((\boldsymbol{T}(\mathbf{w}_a(t))\nabla \boldsymbol{L}(\mathbf{w}_a(t))) \odot \boldsymbol{\Xi}^*(t))]\mathrm{d}t,$$

which completes the proof.

$\square$

**Proposition C.5** (Escaping efficiency of C-SGD). *If the loss $\boldsymbol{L}$ has second-order partial derivatives, the escaping efficiency of C-SGD from minimum $\mathbf{w}^*$ satisfies*

$$\mathbb{E}_{\substack{\mu_j(t)\sim D \\ j=1,\ldots,m}}[\boldsymbol{L}(\mathbf{w}_a(t+1)) - \boldsymbol{L}(\mathbf{w}^*)]$$
$$= - \int_0^t \nabla \boldsymbol{L}(\mathbf{w}_a(t))^T \nabla \boldsymbol{L}(\mathbf{w}_a(t)) + \int_0^t \frac{\eta}{2} \operatorname{Tr}\left(\boldsymbol{H}(\mathbf{w}_a(t))\boldsymbol{\Sigma}_c(t)\right) \mathrm{d}t,$$

*where $\boldsymbol{\Sigma}_c(t)$ denotes the covariance matrix of the gradient noise of C-SGD (Equation (1)).*

The proof is analogous to Theorem 3.

$\square$

**Definition C.1** (Sub-quadratic minimum). *Given that the loss $\boldsymbol{L}$ is continuous and has second-order partial derivatives, we call the mimimum $\mathbf{w}^*$ of $\boldsymbol{L}$ $\delta$-locally sub-quadratic if for any $\mathbf{w}$ in the open punctured neighbourhood of $\mathbf{w}^*$ (i.e., $\mathbf{w} \in \mathring{U}(\mathbf{w}^*)$), the following condition holds: (1) $\boldsymbol{H}(\mathbf{w}^*) \succcurlyeq \boldsymbol{H}(\mathbf{w})$; and (2) $\exists\, \alpha(\mathbf{w}) \in \mathbb{R}^+, \beta(\mathbf{w}) \in \mathbb{R}^-$ s.t. $\boldsymbol{H}(\mathbf{w})(\mathbf{w}-\mathbf{w}^*) = \alpha(\mathbf{w})(\|\mathbf{w}-\mathbf{w}^*\|_2^{\beta(\mathbf{w})}(\mathbf{w}-\mathbf{w}^*))$.*

**Proposition 4.** *$\mathrm{grandsum}((\boldsymbol{T}(\mathbf{w}_a(t))\nabla \boldsymbol{L}(\mathbf{w}_a(t))) \odot \boldsymbol{\Xi}^*(t))$ is (1) zero on quadratic minima, (2) positive on super-quadratic minima, and (3) negative on sub-quadratic minima.*

*Proof of Proposition 4.*

(1) quadratic minima.

It is obvious that on quadratic loss, $\mathrm{grandsum}((\boldsymbol{T}(\mathbf{w}_a(t))\nabla \boldsymbol{L}(\mathbf{w}_a(t))) \odot \boldsymbol{\Xi}^*(t)) = 0$ due to zero gradient diversity (see Corollary C.1).

(2) super-quadratic minima.

Performing the Taylor expansion of $\boldsymbol{H}(\mathbf{w}) - \boldsymbol{H}(\mathbf{w}^*)$ around $\mathbf{w}^*$ provides
$$\boldsymbol{H}(\mathbf{w}) - \boldsymbol{H}(\mathbf{w}^*) \approx \boldsymbol{T}(\mathbf{w})(\mathbf{w} - \mathbf{w}^*) \succcurlyeq 0.$$

According to the definition of super-quadratic minima, we know that $\exists\, \alpha(\mathbf{w}) \in \mathbb{R}^+, \beta(\mathbf{w}) \in \mathbb{R}^+$ s.t.
$$\boldsymbol{T}(\mathbf{w})(\boldsymbol{H}(\mathbf{w})(\mathbf{w} - \mathbf{w}^*)) = \alpha(\mathbf{w})\|\mathbf{w} - \mathbf{w}^*\|_2^{\beta(\mathbf{w})}\boldsymbol{T}(\mathbf{w})(\mathbf{w} - \mathbf{w}^*) \succcurlyeq 0.$$

Another Taylor expansion of $\nabla \boldsymbol{L}(\mathbf{w}) - \nabla \boldsymbol{L}(\mathbf{w}^*)$ around $\mathbf{w}^*$ will give
$$\boldsymbol{T}(\mathbf{w})(\nabla \boldsymbol{L}(\mathbf{w}) - \underbrace{\nabla \boldsymbol{L}(\mathbf{w}^*)}_{0}) \approx \boldsymbol{T}(\mathbf{w})(\boldsymbol{H}(\mathbf{w})(\mathbf{w} - \mathbf{w}^*)) = \alpha(\mathbf{w})\|\mathbf{w} - \mathbf{w}^*\|_2^{\beta(\mathbf{w})}\boldsymbol{T}(\mathbf{w})(\mathbf{w} - \mathbf{w}^*) \succcurlyeq 0.$$

Then we arrive at $\mathrm{grandsum}((\boldsymbol{T}(\mathbf{w}_a(t))\nabla \boldsymbol{L}(\mathbf{w}_a(t))) \odot \boldsymbol{\Xi}^*(t)) > 0$ since $\boldsymbol{\Xi}^*(t)$ is a diagonal matrix with all positive entries.

(3) sub-quadratic minima.

By the same token, we can prove that $\mathrm{grandsum}((\boldsymbol{T}(\mathbf{w}_a(t))\nabla \boldsymbol{L}(\mathbf{w}_a(t))) \odot \boldsymbol{\Xi}^*(t)) < 0$ on sub-quadratic minima.

$\square$

