# OpenReview forum: "Would decentralization hurt generalization?"
_ICLR.cc/2023/Conference — Submitted to ICLR 2023_

### Official Review · Reviewer_hoUP · 2022-10-22

**Confidence:** 4
**Correctness:** 3
**Technical Novelty And Significance:** 2
**Empirical Novelty And Significance:** 2
**Recommendation:** 5

**Clarity, Quality, Novelty And Reproducibility:**

Overall, the quality of the this paper is good. It is easy to follow, but requiring more work to make it technically solid and sound. The clarity of the work should be improved based on the comments provided above. The novelty is also good as the authors tried to address an issue reflected from experimental results and provide new analytical results.

**Strength And Weaknesses:**

I think overall the investigated topic in this paper is quite interesting. Basically the implicit regularization in D-SGD has not sufficiently been investigated based on the empirical results obtained by using D-SGD and C-SGD. Most existing works have paid much attention to the convergence analysis, while ignoring some findings from the experimental results. The authors have showed some new analysis to explain the better generalizability in D-SGD and tried to explain mathematically why this happened. The paper is also easy to follow and well written. However, in the current form, there are quite a few confusions in the paper that require more works from the authors to make it technically solid and sound.

1. It looks like the definition in this paper for C-SGD, namely the centralized SGD, still involves the communication, which is basically the star-network or Federated Learning setting. While this could somehow be confusing. As centralized SGD typically is referred to as a setting that has no local agents or workers, but just only one agent (could be the centralized server) WITHOUT any communication happening during training. Thus, the C-SGD in this work can be more appropriately replaced with "distributed" to avoid confusion. Or some more clarification needs to be detailed in the work.

2. Eq. (1) shows the update for the centralized distributed SGD (C-SGD). It should be noted that from the network topology in Figure 2, communication happens between the server and local agents. The authors should detail this part as well. Distributed learning can involve either just data parallelism, model parallelism or FL. Based on what has been shown in the work, I think it is exactly the FL setting, in which communication would be there between server and agent. Also, there is a typo below Eq.(1) at the bottom of page 3, it should be C-SGD.

3. After Theorem 1, the authors mentioned that "Theorem 1 shows that the decentralization navigates..., in order to lower the regularization term..." How? It is not obviously observed from the theorem statement. The authors should give more detail on this. Additionally, the authors claimed that this is the first work to show the equivalence between D-SGD and C-SGD on a regularized loss with implicit sharpness regularization. Again, the authors should give more technical detail to show instead of just saying that. Where is the regularized loss?

4. How to arrive at Eq.(4)? Where is the gossip matrix \mathbf{P}? It doesn't look like the update of the consensus model for D-SGD. I don't see communication in the formula.

5. The authors mentioned in the paper that the quadratic approximation in the analysis of mini-batch SGD fails to capture how decentralization affects the training dynamics of D-SGD. I am confused here. Based on the statement from the author, then D-SGD performs poor for all scenarios where quadratic losses are there. Note that a lot of regression task is based on mean square error, which is a quadratic loss.

6. The authors repeatedly claimed D-SGD to implicitly regularize the regularization term in the work. It is not obvious for readers, I believe. Can the authors clarify by presenting more detail?

7. How to justify the assumption in Theorem 2? Is it generic in D-SGD? I don't think so.

8. Eq.(6) is obtained by combining Eq.(3) and Eq.(4). How to get that? Substituting Eq.(4) into Eq.(3)?

9. Why did the authors define a new notation in the paper, the so-called escaping efficiency? Such a measure has widely been used in evaluating various stochastic optimization algorithms. Any particular reason to call it so here?

10. In Definition 3, please define all notations necessarily to make the statement clear.

11. For experimental results, though we see the validation for the analysis, to me it is not that promising due to only one dataset. Can the authors add more datasets to validate? Also, I don't think the authors need to impose open problems in the main contents. Instead, they can add them as future directions in the last section. Without any preliminary discussion for these two open problem, that looks a bit weird. For different topologies, the authors have said in the paper that they observed the similar trends. Please add them into the paper to provide more evidence for the theory.

12. Overall, this paper did really try to show some new and good analysis for D-SGD. However, it looks like the span in this work is way too big. Each subsection in Section 4 can be expanded into a standalone paper with comprehensive and thorough theoretical analysis and experimental results. I understand the authors may want to present their findings in time. While with the limit of space, this might hurt the soundness of the work.

******************************Post-rebuttal*****************************
Thanks much for the authors' responses and revisions. I appreciate that. After carefully reviewing the responses from the authors and other reviewers' comments, I will keep my current score. Though additional changes have been made to the draft, it still didn't look technically sound and solid. Particularly, the empirical evidences to validate the theory are still weak, and some analysis in the work still requires more clarification.

**Summary Of The Paper:**

This paper investigates the existing conclusion that decentralized stochastic gradient descent (D-SGD) degrades the generalizability, which conflicts with experimental results in large-batch settings that D-SGD generalizes better than centralized distributed SGD (C-SGD). Specifically, the authors present new theoretical analysis to reconciles the conflict. They show that D-SGD introduces an implicit regularization to enable the escaping of the sharpness of the learned minima. Additionally, they theoretically prove that the implicit regularization is amplified in large-batch settings when the linear scaling rule is used. In their work, they also show that D-SGD converges to super-quadratic flat minima eventually. To validate the analysis presented in the paper, the authors utilizes a benchmark dataset with a few models to show empirical results, compared to C-SGD.

**Summary Of The Review:**

This paper investigates the existing conclusion that decentralized stochastic gradient descent (D-SGD) degrades the generalizability, which conflicts with experimental results in large-batch settings that D-SGD generalizes better than centralized distributed SGD (C-SGD). The authors provided theoretical analysis for the implicit regularization in D-SGD and showed that D-SGD can avoid the sharpness of the learned minima. They also presented some preliminary experimental result to support their findings. The paper looks good, but more work is required for clarity and novelty.

---

> ### Author Response · Authors · 2022-11-28
> **To Reviewer hoUP**
>
> Thank you for recognizing our paper's quality, and we appreciate your insightful comments. All your concerns have been carefully responded to below. The manuscript is carefully revised according to your suggestions. We sincerely hope our responses fully address your concerns.
>
> > **Q1**: C-SGD in this work can be more appropriately replaced with "distributed" to avoid confusion.
>
> **A1**: Thanks. The word "centralized" indicates that C-SGD has a central server. We will note this in the manuscript.
>
> > **Q2**: Is equation (1) the FL setting?
>
> **A2**: Yes, it is. We will note this in the manuscript.
>
> > **Q3**: Where is the regularized loss? Give some details about the implicit sharpness regularization.
>
> **A3**: Thanks. the regularized loss is as follows,
>
> $$\boldsymbol{L}\left(\mathbf{w}_a(t)\right)+\frac{1}{2} \operatorname{Tr}\left(\boldsymbol{H}\left(\mathbf{w}_a(t)\right) \boldsymbol{\Xi}^*(t)\right).$$
>
> Meanwhile, the objective function in C-SGD is as follows,
>
> $$\mathbb{E}_{\substack{\mu_j(t) \sim D \\ j=1, \ldots, m}}\left[\mathbf{w}_a(t+1)\right]=\mathbf{w}_a(t)-\eta \nabla \boldsymbol{L}\left(\mathbf{w}_a(t)\right).$$
>
> Comparing them, omitting higher-order residual terms, we find D-SGD has additional terms as follows,
>
> $$\frac{1}{2} \operatorname{Tr}\left(\boldsymbol{H}\left(\mathbf{w}_a(t)\right) \boldsymbol{\Xi}^*(t)\right) +\mathcal{O}\left(\eta^2 \mathbf{1}\right).$$
>
> The second term $\mathcal{O}\left(\eta^2 \mathbf{1}\right)$ will become negligible compared with the gradient related term $-\eta \nabla\left[\boldsymbol{L}\left(\mathbf{w}_a(t)\right)+\frac{1}{2} \operatorname{Tr}\left(\boldsymbol{H}\left(\mathbf{w}_a(t)\right) \boldsymbol{\Xi}^*(t)\right)\right]=\mathcal{O}\left(\eta \mathbf{1}\right)$ if $\eta$ is sufficiently small. $\frac{1}{2} \operatorname{Tr}\left(\boldsymbol{H}\left(\mathbf{w}_a(t)\right) \boldsymbol{\Xi}^*(t)\right)$ is exactly the regularization term. Please also kindly refer to Theorem 1.
>
> > **Q4**: How to arrive at Eq.(4)? Where is the gossip matrix $\mathbf{P}$?
>
> **A4**:  Thanks. According to the fact that $\frac{1}{m}\sum_{j=1}^m\sum_{j=1}^m \mathbf{P}_{j, k} \mathbf{w}_k(t)=\mathbf{w}_a(t)$, averaging over all workers $j$ in equation (2) gives
>
> $$
> w_a(t+1)=w_a(t)-\frac{1}{m}\sum_{j=1}^{m}\eta \cdot \nabla \boldsymbol{L}^{\mu_j(t)}\left(w_j(t)\right).
> $$
> Then we arrive at
>
> $$w_{a}(t+1)=w_{a}(t)-\eta\left[\nabla \boldsymbol{L}\left(w_a(t)\right)+ [\nabla \boldsymbol{L}^{\mu(t)}\left(w_a(t)\right)-\nabla \boldsymbol{L}\left(w_a(t)\right)] +\frac{1}{m}\sum_{j=1}^{m}[\nabla \boldsymbol{L}^{\mu_j(t)}\left(w_j(t)\right) - \nabla \boldsymbol{L}^{\mu(t)}\left(w_a(t)\right)]\right].$$
>
> We sincerely note that the global averaged model $w_a{(t)}$ in equation (4) does depend on the gossip matrix $\mathbf{P}$.
>
> > **Q5**: How to justify the assumption in Theorem 2? Is it generic in D-SGD?
>
> **A5**: Thanks. Theorem 2 follows the consensus distance theory in (Kong et al., 2021) (see Lemma 3 on page 17). It assumes that the gradient norm does not decrease too fast.
>
> > **Q6**: What does the author mean by saying, "the quadratic approximation in the analysis of mini-batch SGD fails to capture how decentralization affects the training dynamics of D-SGD"?
>
> **A6**: We mean that under quadratic approximation, the training dynamics of D-SGD are exactly the same as that of C-SGD (see proof in Corollary C.1). Therefore, the quadratic approximation cannot capture the difference between D-SGD and C-SGD. To address this problem, we perform a higher Taylor expansion on the gradient diversity around $\mathbf{w}_a(t)$. Please kindly refer to Section 4.1.
>
> > **Q7**: Why did the authors define a new notation in the paper, the so-called escaping efficiency?
>
> **A7**: We sincerely note that this notion has been studied in many existing works (Zhu et al., 2019; Liu et al., 2021). Intuitively, this notion characterizes how hard it is for D-SGD to "escape" if it gets stuck in a minimum $\mathbf{w}^*$.
>
> > Q8: Please define all notations in Definition 3.
>
> **A8**: Thanks and addressed. Please also kindly refer to our manuscript.
>
> > **Q9**: Please add experiments on different topologies.
>
> **A9**: Thanks. In the rebuttal, we conducted additional experiments, which currently cover mesh-grid and static exponential topologies. The added empirical results are presented at [https://github.com/anonymousforrebuttal/anonymous_code](https://github.com/anonymousforrebuttal/anonymous_code)..
>
> > **Q10**: Please add the open problems in Section 6 as future directions in the last section.
>
> A10: Thank you for your suggestion! We have revised the manuscript accordingly.
>
> Reference:
>
> [1] Zhu et al. "The anisotropic noise in stochastic gradient descent." ICML 2018.
>
> [2] Kong et al. "Consensus control for decentralized deep learning." ICML 2020.

---

### Official Review · Reviewer_m7pE · 2022-10-22

**Confidence:** 3
**Correctness:** 2
**Technical Novelty And Significance:** 2
**Empirical Novelty And Significance:** 2
**Recommendation:** 3

**Clarity, Quality, Novelty And Reproducibility:**

The paper is of poor quality, and as I mentioned previously, some results and proofs are non-rigorous. The topic is worth investigating and some ideas are novel, but the paper is poorly executed. I have some further comments as follows.

(1) In Proposition C.5., should it be D-SGD instead of C-SGD?

(2) In references, some letters need to be capitalized. For example, Entropy-sgd should be Entropy-SGD in Chaudhari et al., (2019) schur should be Schur in Davis (1962), and sgd should be SGD in Goyal et al. (2017).

(3) In Theorem 1, because you consider the average iterates, instead of individual iterates, the doubly stochastic matrix does not seem to play a role? How does the network structure affect your main result?

**Strength And Weaknesses:**

In my view, the paper studies a very important topic. Whether decentralization can help or hurt generalization is a very interesting research topic. The authors' idea to show that the decentralized SGD can introduce an implicit regularization, which might help generalization is a novel idea, and has some potential.

However, I am not satisfied with the overall quality of the paper.

First of all, there is little technical novelty or contribution here. The authors only looked at the average of iterates, instead of the individual iterates, and as a result, when you average out over the doubly stochastic matrix, it disappears from the analysis, and makes the analysis easy. However, it is well known that the average iterates are close to the individual iterates (see e.g. Yuan et al. "On the convergence of decentralized gradient descent" for the deterministic gradient case and Fallah et al. "Robust distributed accelerated stochastic gradient methods for multi-agent networks" for the stochastic gradient case). It would be nice if the authors can obtain some results for individual iterates instead of the average iterates.

Second, some conclusion and implication from the main result are not convincing to me. For example, in Theorem 1, the authors claim that D-SGD implicitly regularizes $\lambda_{H(w_{a}(t)),1}\cdot\text{Tr}(\Xi(t))$. However, if you look at the proof, this term is only an upper bound. It is not very convincing to me. Also, Theorem 1 looks like writing D-SGD as centralized SGD plus some additional term, and this idea is not new, see e.g. Fallah et al. "Robust distributed accelerated stochastic gradient methods for multi-agent networks".

What's even more disappointing is that I am not 100 percent sure that the main result, i.e. Theorem 1 is correct. The proof is not rigorous. For example, in the proof of Theorem 1 in the Appendix, the authors did not explain how to control the remainder term R. What's even more troublesome to me is that in the proof, the authors wrote that "Assuming for the sake of intuition, we expect each dimension of $(w_{j}(\tau)-w_{a}(\tau))$ to be uncorrelated." Why? Please explain. This is not clear to me at all. You cannot simply assume something that you do not know how to prove. Also in the proof, the authors wrote that "if the topology is symmetric". Please explain what you meant by saying "if the topology is symmetric". It seems the authors lack mathematical rigor and basic training in mathematics. In the statement Lemma C.4., the authors wrote that "D-SGD can be viewed as the discretization of the following SDE..." This is not a rigorous mathematical statement at all. It is okay to write this sentence as an informal discussion within a paragraph in the main body of the paper, but one cannot write a statement like that in a lemma. If you want to state it in a lemma, you should make the statement formal and rigorous. For example, see Li et al. "Stochastic modified equations and adaptive stochastic gradient algorithms", where they used weak convergence of certain order to describe SGD being approximated by an SDE.

Finally, there is a disconnection of Theorem 1 and the other results in the paper, like escaping efficiency in Theorem 3. You should make the connections more clear.

**Summary Of The Paper:**

The paper studies whether decentralized stochastic gradient descent (D-SGD) can lead to better generalization compared to the centralized SGD. The authors showed that D-SGD implies an implicit regularization and used this result (Theorem 1) to argue that the implicit regularization promotes the generalization. The authors also study the escaping efficiency of D-SGD and showed that D-SGD favors the so-called super-quadratic flat minima.

**Summary Of The Review:**

The paper studies an important and interesting topic. But I have doubt about the mathematical rigor and main results in the paper.

---

> ### Author Response · Authors · 2022-11-28
> **To reviewer m7pE**
>
> Thank you for your thorough and constructive review. All your questions have been carefully responded to below. We sincerely hope our responses fully address your concerns.
>
> > **Q1**: The authors only looked at the average of iterates, instead of the individual iterates. Averaging out over the doubly stochastic matrix makes the analysis easy.
>
> **A1**: Thanks. Thanks. Since the average iterates are close to the individual iterates (Yuan et al., 2016), averaging out over the doubly stochastic matrix is a novel and effective way to study the training dynamics of D-SGD. Furthermore, the effect of communication is still incorporated in the last term in the RHS of equation (4) (i.e., the gradient diversity among workers).
>
> > **Q2**: The idea of writing D-SGD as centralized SGD plus some additional term has been previously studied in Fallah et al. "Robust distributed accelerated stochastic gradient methods for multi-agent networks".
>
> **A2**: Thanks for pointing this out. Fallah et al. (2019) prove that D-GD optimizes $F_{\mathbf{P}, \alpha}(w):=\frac{1}{2 \alpha} w^T\left(I_{N d}-\mathbf{P}\right) w+F(w)$. The major difference between our work and Fallah et al. (2019) is that we study the implicit sharpness regularization of D-SGD, but they do not analyze how the additional term impact the generalizability of D-SGD. We add the discussion in the revision.
>
> > **Q3**: About the rigorousness of proof.
>
> **A3**: Thanks. We have revised the proof. Please kindly refer to Appendix C.
>
> > **Q4**:  Discuss the connection between Theorem 1 and the other results in the paper, like escaping efficiency in Theorem 3.
>
> **A4**: Thanks for your suggestion. We want to analyze the generalizability of D-SGD from two perspectives. Theorem 1 proves the loss function that D-SGD implicitly optimizes. Theorem 2 studies the escaping efficiency of D-SGD, which has been previously studied in the literature to explore the generalizability of SGD (Zhu et al., 2018).
> The escaping efficiency
> $\mathbb{E}_{\mathbf{w}_a(t)}\left[\boldsymbol{L}\left(\mathbf{w}_a(t)\right)-\boldsymbol{L}\left(\mathbf{w}^*\right)\right]$
> characterizes how hard it is for D-SGD to "escape" if it gets stuck in a minimum w star.
>
> Reference:
>
> [1] Yuan et al. "On the convergence of decentralized gradient descent" for the deterministic gradient case and Fallah et al." SIAM 2016.
>
> [2] Zhu et al. "The anisotropic noise in stochastic gradient descent." ICML 2018.

---

> ### Author Response · Authors · 2022-12-08
> **Sincerely looking forward to your reply**
>
> Dear Reviewer m7pE,
>
> Thank you very much for your thorough comments and helpful comments. We are looking forward to your reply. If you have further concerns or requests, please kindly give us a chance to address them in the discussion period. If all your concerns have been resolved, it is much appreciated if you may raise the rating of our work.
>
> Thank you!
>
> The authors

---

### Official Review · Reviewer_baeM · 2022-10-28

**Confidence:** 3
**Correctness:** 1
**Technical Novelty And Significance:** 1
**Empirical Novelty And Significance:** 2
**Recommendation:** 1

**Clarity, Quality, Novelty And Reproducibility:**

The paper is also poorly written and unclear: many notions are not defined clearly. For example, I cannot find the definitions of H and T in the main text, which is an egregious omission since it is not standard notation. Some other comments, typos, etc.:
- Pg. 1, “decentralization nature” -> “decentralized nature”
- Pg. 1, “This conflict signifies that the major characteristics…” What major characteristics?
- Pg. 1, paragraph after the box is redundant.
- Last bullet point in section 1, “continuou-time SGD”
- Section 2: “flat minimum varies slowly” No, a flat minimum does not vary; the loss around a flat minimum varies.
- Underneath eq. (1), what does it mean that ζ is a random variable? From the equation it seems that ζ is a summation index?
- Top of pg. 4, “equals to the” -> “equals the”
- Eq. (2), extra comma
- Section 4, “minimizes the sharpness of Hessian” -> remove the words “of Hessian”
- Section 4.1, rename section to “D-SGD is equivalent to C-SGD on a regularized loss” and edit the subsequent paragraph accordingly
- Thm. 1, what does it mean that the probability is greater than 1 - O(η)? Where is the randomness coming over? On the LHS of the equality there is already an expectation
- Pg. 4, inconsistent notation: is the subscript of the maximum Hessian eigenvalue 1 or max?
- Pg. 6, missing * for the consensus distance
- Defn. 3, U is not defined
- Pg. 9, what is a “spare” topology?


**Strength And Weaknesses:**

The strength of the paper is that it tackles a relevant problem (generalization in decentralized settings) and makes an interesting claim about the solutions that D-SGD finds.

The main weakness of the paper is that the argument presented therein is sketchy and not at all convincing. I have the following questions:
- In eq. (3), are we considering the regime where the step size η is large or small? If η is small, then wouldn’t the O(η^{½}) term dominate the main “regularized loss” term? In any case, the last term also seems to be of order O(η), so why is it not considered part of the “regularized loss”? It is completely unclear why we ignore certain parts of eq. (3) and focus on others, and **this point alone completely invalidates the reasoning of the paper**.
- In step (3) at the end of section 4.1, how is the upper inequality established? Please provide a proof.
- In the SDE after eq. (6), why does the noise involve η? When deriving an SDE limit, aren’t we taking η to zero?
- The definition of escaping efficiency does not capture the notion of escaping at all. First of all, the definition of escaping efficiency only talks about the value of the loss, it does not talk about whether or not we “escape” (i.e., travel away from) w^*. Second, the escaping efficiency does not “characterize” a probability, because Markov’s inequality is not guaranteed to be tight. Third, what is the interpretation of escaping efficiency? Is a large escaping efficiency supposed to mean that we are more likely to escape? If so, Prop. 4 seems to indicate that D-SGD escapes from super-quadratic minima, which is the opposite conclusion as asserted in the text.

Essentially this paper makes a lot of claims based on extremely shoddy or missing reasoning, which makes this paper **not rigorous**.

**Summary Of The Paper:**

This paper studies decentralized stochastic gradient descent (D-SGD), in which different nodes have different subsets of the training data and communicate with their neighbors (through a graph structure) in order to minimize a loss function. The paper makes the case that D-SGD has an implicit regularization effect that causes it to minimize the sharpness (largest eigenvalue of the Hessian of the loss) and therefore prefer flatter minima.

**Summary Of The Review:**

In summary I found the paper to be very unclear, and it based upon unclear reasoning. Hence I recommend rejection.

---

> ### Author Response · Authors · 2022-11-28
> **To reviewer baeM**
>
> > **Q5**: About the definition and the interpretation of escaping efficiency. The definition of escaping efficiency only talks about the value of the loss, it does not talk about whether or not we "escape" (i.e., travel away from) $w^*$.
>
> **A5**: Thanks. We agree with the reviewer that the definition of escaping efficiency cannot fully capture whether or not we "escape" (i.e., travel away from) $w^*$.
>
> The escaping efficiency measures how far $w$ travels away from $w^*$ in function space rather than in weight space.
> Intuitively speaking, the escaping efficiency $\mathbb{E}_{\mathbf{w}_a(t)}\left[\boldsymbol{L}\left(\mathbf{w}_a(t)\right)-\boldsymbol{L}\left(\mathbf{w}^*\right)\right]$ characterizes how hard it is for D-SGD to "escape" if it gets stuck in a minimum w star.
>
> We prove that $\forall\delta$,
>
> $$P(L(w_{a}{\scriptstyle (t+1)})-L(w^*)\geq \delta)\leq \left[-\int_0^t \mathbb{E}_{\mathbf{w}_a(t)}\left[\nabla \boldsymbol{L}\left(\mathbf{w}_a(t)\right)^T \nabla \boldsymbol{L}\left(\mathbf{w}_a(t)\right)-\frac{1}{2} \operatorname{grandsum}\left(\left(\boldsymbol{T}\left(\mathbf{w}_a(t)\right) \nabla \boldsymbol{L}\left(\mathbf{w}_a(t)\right)\right) \odot \boldsymbol{\Xi}^*(t)\right)\right] \mathrm{d} t+\int_0^t \frac{\eta}{2} \operatorname{Tr}\left(\boldsymbol{H}\left(\mathbf{w}_a(t)\right) \boldsymbol{\Sigma}_D(t)\right) \mathrm{d} t\right]/\delta.$$
>
> > **Q6**: Is a large escaping efficiency supposed to mean that we are more likely to escape?
>
> **A6**: Yes! Intuitively, escaping efficiency measures how hard it is for D-SGD to "escape" if it gets stuck in a minimum $w^*$xxx. Thus, a large escaping efficiency is supposed to mean that we are more likely to escape. There is a **typo** in theorem 3. It should be $-\frac{1}{2} \operatorname{grandsum}\left(\left(\boldsymbol{T}\left(\mathbf{w}_a(t)\right) \nabla \boldsymbol{L}\left(\mathbf{w}_a(t)\right)\right) \odot \boldsymbol{\Xi}^*(t)\right) \mathrm{d} t$ rather than $+\frac{1}{2} \operatorname{grandsum}\left(\left(\boldsymbol{T}\left(\mathbf{w}_a(t)\right) \nabla \boldsymbol{L}\left(\mathbf{w}_a(t)\right)\right) \odot \boldsymbol{\Xi}^*(t)\right) \mathrm{d} t$. In this way, proposition 4 is consistent with our main conclusion. We sincerely apologize for this! We have revised the manuscript accordingly.
>
> > **Q7**: What is a "spare" topology?
>
> **A7**: Thanks. The "sparsity" is measured by the spectral gap. A sparse topology has a small spectral gap. Please kindly refer to Definition A.2 in Appendix A.
>
> > **Q8**: Typos.
>
> **A8**: Thanks and addressed.
>
> Reference:
>
> [1] Smith and Le. "A bayesian perspective on generalization and stochastic gradient descent." ICLR 2018.
>
> [2] Chaudhari and Soatto. "Stochastic gradient descent performs variational inference, con-verges to limit cycles for deep networks." ICLR 2018.
>
> [3] Şimşekli et al. "On the heavy-tailed theory of stochastic gradient descent for deep neural networks." arxiv 2019.
>
> [4] Shi et al. "On learning rates and schrodinger operators." arxiv 2020.

---

> > ### Comment · Reviewer_baeM · 2022-12-08
> > **Response**
> >
> > Thank you for your response. After reading it, however, there are still concerns in my mind (the claimed regularization effect is just a bound on the true regularization effect; no rigorous justification is given for why the cubic term is small, etc.) and the substantial issues present in the first draft of the work lead me to believe that it is not suitable for publication at ICLR.

---

> ### Author Response · Authors · 2022-11-28
> **To reviewer baeM**
>
> Thank you for your thorough and constructive review. All your questions have been carefully responded to below. We sincerely hope our responses fully address your concerns. Very much appreciated!
>
> > **Q1**: In theorem 1, if η is small, then wouldn't the O(η^{½}) term dominate the main "regularized loss" term? In any case, the last term also seems to be of order O(η), so why is it not considered part of the "regularized loss"?
>
> **A1**: Thanks for this insightful question! The last term $\mathcal{O}(\eta\cdot \frac{1}{m}\sum_{j=1}^m\|\|w_j(t)-w_a(t)\|\|_2^3$ denotes the high order residual terms in the Taylor expansion.
>
> This term is of the order $\mathcal{O}(\eta)$ but will be much closer to $0$ than the regularized loss term(which is order $\mathcal{O}(\eta\cdot\frac{1}{m}\sum_{j=1}^m\|\|w_j(t)-w_a(t)\|\|_2^2)$
> when the local models $w_j(t)$ approaches the global averaged model $w_a(t)$.
> For the $\mathcal{O}(\eta^{0.5})$ term, we have improved it to order $\mathcal{O}(\eta^{2})$. Consequently, the "regularized loss" term dominates the training dynamics of D-SGD for small $\eta$.
>
> > **Q2**: Thm. 1, what does it mean that the probability is greater than 1 - O(η)? Where is the randomness coming over? On the LHS of the equality, there is already an expectation.
>
> **A2**: Thanks. The additional gradient diversity term $\frac{1}{m} \sum_{j=1}^m\left[\nabla \boldsymbol{L}^{\mu_j(t)}\left(\mathbf{w}_j(t)\right)-\nabla \boldsymbol{L}^{\mu_j(t)}\left(\mathbf{w}_a(t)\right)\right]$ (see equation (4)) depends on the random data drawn from 1-th to $(t-1)$-th iteration.
>
> This introduces randomness over $\mathbb{E}_{\substack{\mu_j(t) \sim D \\ j=1, \ldots, m}}\left[\mathbf{w}_a(t+1)\right]$.
>
> > **Q3**: In step (3) at the end of section 4.1, how is the upper inequality established?
>
> **A3**: Thanks. We have already provided the proof. The intuition is that when $\frac{\lambda_{H\left(w_a(t)\right), 1}}{\textrm{Tr}\left(H\left(w_a(t)\right)\right)}$ is small,
> $\textrm{Tr}\left(\Xi_{(t)}\right)\cdot\lambda_{H\left(w_a(t)\right), 1}$ will be close to $\textrm{Tr}\left(H\left(w_a(t)\right) \Xi^*(t)\right)$. Please kindly refer to Lemma C.2 in Appendix C .
>
> > **Q4**: In the SDE after eq. (6), why does the noise involve η? When deriving an SDE limit, aren't we taking η to zero?
>
> **A4**: Thanks. Retaining both $\mathcal{O}(\eta)$ and $\mathcal{O}(1)$ terms and omitting the $\mathcal{O}\left(\eta^{2} \mathbf{1}\right)$ term as well as the higher-order residual terms, equation (6) provides
>
> $$
> \mathbf{w}_a(t+1)
> =\mathbf{w}_a(t)-\eta \nabla\left[\boldsymbol{L}\left(\mathbf{w}_a(t)\right)+\frac{1}{2} \operatorname{Tr}\left(\boldsymbol{H}\left(\mathbf{w}_a(t)\right) \boldsymbol{\Xi}^*(t)\right)\right]+\eta \epsilon^0(t),
> $$
>
> where $\epsilon^0{\scriptstyle (t)}$ denotes the zero-mean noise in D-SGD. Absorbing $\eta$ guarantees
>
> $$\mathrm{d} \mathbf{w}_a(t)=-\left[\nabla \boldsymbol{L}\left(\mathbf{w}_a(t)\right)+\frac{1}{2} \boldsymbol{T}\left(\mathbf{w}_a(t)\right)  \boldsymbol{\Xi}^*(t)-\epsilon^0(t)\right] \mathrm{d} t.$$
>
> We denote $W$ as a standard Brownian motion and assume $\epsilon^0(t)$ follows a Gaussian distribution whose covariance matrix is $\boldsymbol{\Sigma}_{\mathrm{D}}(t)$. Then we have
>
> $$\sqrt{\eta}\epsilon^0(t)=\sqrt{\Sigma_{\mathrm{D}}(t)}(W_{(t+1)}-W_{(t)})=\eta\sqrt{\boldsymbol{\Sigma}_{\mathrm{D}}(t)} \frac{\mathrm{d} W\left(t\right)}{\mathrm{d} t}+O\left(\eta^2\right),$$
>
> which gives
>
> $$\epsilon^0(t)=\sqrt{\eta \boldsymbol{\Sigma}_{\mathrm{D}}(t)} \frac{\mathrm{d} W\left(t\right)}{\mathrm{d} t}+O\left(\eta^{\frac{3}{2}}\right).$$
>
> Plugging the display into the above SDE and ignoring smaller $O\left(\eta^{1.5}\right)$ terms, we arrive at
>
> $$\mathrm{d} w_a(t)=-\left[\nabla \boldsymbol{L}\left(w_a(t)\right)+\frac{1}{2} T\left(w_a(t)\right)  \Xi^*(t)\right] \mathrm{d}t+\sqrt{\eta \Sigma_{\mathrm{D}}(t)} \mathrm{d} W(t).$$
>
> This technique has been widely used in the existing literature (Smith and Le, 2018; Chaudhari and Soatto, 2018; Şimşekli et al., 2019; Shi et al., 2020).

---

> ### Author Response · Authors · 2022-12-08
> **Sincerely looking forward to your reply**
>
> Dear Reviewer baeM,
>
> Thank you very much for your thorough comments and helpful comments. We are looking forward to your reply. If you have further concerns or requests, please kindly give us a chance to address them in the discussion period. If all your concerns have been resolved, it is much appreciated if you may raise the rating of our work.
>
> Thank you!
>
> The authors

---

### Official Review · Reviewer_qFYy · 2022-10-28

**Confidence:** 2
**Clarity, Quality, Novelty And Reproducibility:** 1. Should remove "The code will be re…
**Correctness:** 2
**Technical Novelty And Significance:** 3
**Empirical Novelty And Significance:** 3
**Recommendation:** 5

**Strength And Weaknesses:**

The topic is very interesting and it has important implications for understanding the impact of decentralized algorithms on optimization problems. However the main result (Theorem 1) confuses me, since it says with high probability, the expectation of w(t+1) = w(t) - \eta * gradient & regularization term + O(\sqrt{\eta}) + O(\eta * weight diversity related term). This does not seem to make sense because when \eta (which is the step length) becomes very small, the result seems to indicate w(t) will diverse and the gradient related term becomes minor compared to the other terms. This contradicts with our common intuition that a smaller step size leads to easier convergence.

**Summary Of The Paper:**

The authors did theoretical analysis for the D-SGD algorithm and the results are claimed to show an implicit regularization during D-SGD optimization process that penalizes the learned minima’s sharpness. The escaping efficiency of the D-SGD algorithm is also analyzed.

**Summary Of The Review:**

See section "Strength And Weaknesses"

---

> ### Author Response · Authors · 2022-11-28
> **To reviewer qFYy**
>
> Thank you for your recognizing the significance of our contribution; we appreciate your constructive feedback. All your questions have been carefully responded to below. We sincerely hope our responses fully address your concerns.
>
> > **Q1**: Theorem 1 implies that D-SGD will diverge if the learning rate is small.
>
> **A1**: Thanks. We have redressed this in the rebuttal. We prove that,
>
> $$\mathbb{E}_{\substack{\mu_j(t) \sim D \\ j=1, \ldots, m}}\left[\mathbf{w}_a(t+1)\right]=\mathbf{w}_a(t)-\eta \nabla\left[\boldsymbol{L}\left(\mathbf{w}_a(t)\right)+\frac{1}{2} \operatorname{Tr}\left(\boldsymbol{H}\left(\mathbf{w}_a(t)\right) \boldsymbol{\Xi}^*(t)\right)\right]+\mathcal{O}\left(\eta^2 \mathbf{1}\right)+\mathcal{O}\left(\eta\left\|\|\mathbf{w}_j(t)-\mathbf{w}_a(t)\right\|\|_2^3 \mathbf{1}\right).$$
>
> The $\mathcal{O}\left(\eta^2 \mathbf{1}\right)$ term will become negligible compared with the gradient related term $-\eta \nabla\left[\boldsymbol{L}\left(\mathbf{w}_a(t)\right)+\frac{1}{2} \operatorname{Tr}\left(\boldsymbol{H}\left(\mathbf{w}_a(t)\right) \boldsymbol{\Xi}^*(t)\right)\right]=\mathcal{O}\left(\eta \mathbf{1}\right)$ if $\eta$ is sufficiently small. The high order residual term $\mathcal{O}\left(\eta\left\|\|\mathbf{w}_j(t)-\mathbf{w}_a(t)\right\|\|_2^3 \mathbf{1}\right)$ will converges to zero faster than $\frac{1}{2}\eta \nabla\left[\operatorname{Tr}\left(\boldsymbol{H}\left(\mathbf{w}_a(t)\right) \boldsymbol{\Xi}^*(t)\right)\right] = \mathcal{O}\left(\eta\left\|\|\mathbf{w}_j(t)-\mathbf{w}_a(t)\right\|\|_2^2 \mathbf{1}\right)$ when the local models $\mathbf{w}_j(t)$ approaches the global averaged model $\mathbf{w}_a(t)$. Consequently, the result show that $\mathbf{w}_a(t)$ will not diverge since the gradient related term $-\eta \nabla\left[\boldsymbol{L}\left(\mathbf{w}_a(t)\right)+\frac{1}{2} \operatorname{Tr}\left(\boldsymbol{H}\left(\mathbf{w}_a(t)\right) \boldsymbol{\Xi}^*(t)\right)\right]$ will gradually dominate the other terms during training.
>
> > **Q2**: Please put a link to the code.
>
> **A2**: Thanks. The code is provided at https://github.com/anonymousforrebuttal/anonymous_code.

---

### Official Review · Reviewer_Phej · 2022-10-29

**Confidence:** 3
**Correctness:** 2
**Technical Novelty And Significance:** 2
**Empirical Novelty And Significance:** Not applicable
**Recommendation:** 3

**Clarity, Quality, Novelty And Reproducibility:**

Clarity: The definitions are clearly stated. Are these the standard ones? Also, the proofs presented in the appendix are not rigorous. Detailed explanations are encouraged to add.

Quality: this paper only introduces the idea of recognizing the consensus violation as an implicit regularization but does not quantify that the generalization error has indeed a direct causal effect with this term.

Novelty: to my knowledge, this way of looking at the decentralized generalization performance is new.

Reproducibility: no detailed hyperparameters are provided either in the main text or appendix. The code is not attached. There is no way to reproduce it.



**Strength And Weaknesses:**

The major strength of this work is studying the generalization performance of decentralized SGD compared with the centralized case by analyzing the eigenvalues of the hessian. The technical contributions include implicit regularization analysis, amplified regularization by large batches, escaping efficiency, and numerical evaluations.

The most concerns of this work are as follows:

1) This is straightforward to see that the consensus violation would serve as the implicit regularization during the training phase, but how much this play on the generalization error is not analyzed. There is a gap between this observation and generalization performance.

2) The large batch amplification is not clear. On page 20, it said that the regularization coefficient is $\eta\textrm{Tr}$. Please let me know when it was mentioned in the Thm.1. Note that the consensus error defined in this work is not dependent on the step size. Also, even it was true, $\eta^3$ is overclaimed as at least the size of the gradient of the original loss is at least proportional to $\eta^2$. Again, this issue originated from which metric the authors want to compare, weight, loss function, gradient, or generalization error.

3) Even more problematic is that def.1 is rather weak. Is it from a theoretical analysis or just empirical observation? either one is fine, but here it serves as a definition. Assume this was a definition. It said a fixed learning rate to batch size ratio. From the equation shown in the proof of thm 2 (page 20. please use equation indices), there is no batch size. Def. 1 does not imply that here the learning rate can be replaced by the batch size. Also, the larger the $\eta$ is, the less likely the thm 1. holds as claimed by the authors on page 6.

4) It does not make sense to have a $w^*$ here as the authors consider the nonconvex problems. There is no clue that DSGD can converge to this point, even it is a local minimum.




**Summary Of The Paper:**

This paper studies the generalization performance of decentralized training compared with the centralized one, especially for the large batch setting. The authors show that there is an implicit regularization that penalizes the sharpness of the learned minima and the consensus violation, and this regularization is amplified by the large batch size. Also, the authors claim that decentralized SGD is more likely to find the super quadratic local minima.

**Summary Of The Review:**

In summary, this paper considers a critical issue of addressing the generalization performance of decentralized SGD, however, the provided argument is not direct linking the consensus violation and generalization error. The theoretical argument is not rigorously justified even with good intuitions.

---

> ### Author Response · Authors · 2022-11-28
> **To reviewer Phej**
>
> Reference:
>
> [1] Sepp Hochreiter and Jurgen Schmidhuber. "Flat minima." Neural computation, 1997.
>
> [2] Jiang et al. "Fantastic generalization measures and where to find them." ICLR 2020.
>
> [3] He et al. "Deep residual learning for image recognition." CVPR 2016.
>
> [4] Goyal et al. "Accurate, large minibatch SGD: Training ImageNet in 1 hour." arxiv 2017.
>
> [5] Smith et al. "On the generalization benefit of noise in stochastic gradient descent." ICML 2020.
>
> [6] Li et al. "On the Validity of Modeling SGD with Stochastic Differential Equations (SDEs)." NeurIPS 2021.

---

> ### Author Response · Authors · 2022-11-28
> **To reviewer Phej**
>
> Thank you for recognizing the clarity and novelty of our paper; we appreciate your constructive comments. All your concerns have been carefully responded to below. The manuscript is carefully revised according to your suggestions. We sincerely hope our responses fully address your concerns.
>
> > **Q1**: Which metric of C-SGD and D-SGD do the authors want to compare (weight, loss function, gradient, or generalization error)?
>
> **A1**: Thanks. Technically, we compare the objective functions in C-SGD and D-SGD, which shows that the objective function in D-SGD has an additional regularization term. This term introduces an implicit regularization on the trace of Hessian. As far as the concern on generalization, the advantage of decentralization is measured by the comparison of generalization error.
>
> > **Q2**: The relationship (and causal effect) between the implicit sharpness regularization term and the generalization error.
>
> **A2**: Thanks. The flatness of minimum has long been regarded as a proxy of generalization (Hochreiter & Schmidhuber, 1997; Jiang et al., 2020). Theorem 1 states that D-SGD optimizes the original loss function with an additional sharpness regularization term. The Theorem then implies that the solutions found by D-SGD are more likely to be wider on the landscape than these found by C-SGD and thus D-SGD have the potential to generalize better. A more direct causal effect can be obtained by constructing tight sharpness-aware generalization bounds of D-SGD, which could be a potential future work.
>
>  > **Q3**: The large batch amplification is not clear. Why the regularization coefficient is $\eta \operatorname{Tr}\left(\boldsymbol{\Xi}_{(t)}\right)$?
>
> **A3**: Thanks. We prove that the objective function in D-SGD is as follows,
>
> $$\mathbb{E}_{\substack{\mu_j(t) \sim D \\ j=1, \ldots, m}}\left[\mathbf{w}_a(t+1)\right]=\mathbf{w}_a(t)-\eta \nabla \left[\boldsymbol{L}\left(\mathbf{w}_a(t)\right)+\frac{1}{2} \textrm{Tr}\left(\boldsymbol{H}\left(\mathbf{w}_a(t)\right) \boldsymbol{\Xi}^*(t)\right)\right]+\mathcal{O}\left(\eta^2 \mathbf{1}\right)+\mathcal{O}\left(\eta\left\|\|\mathbf{w}_j(t)-\mathbf{w}_a(t)\right\|\|_2^3 \mathbf{1}\right).$$
>
> Meanwhile, the objective function in C-SGD is as follows,
>
> $$\mathbb{E}_{\substack{\mu_j(t) \sim D \\ j=1, \ldots, m}}\left[\mathbf{w}_a(t+1)\right]=\mathbf{w}_a(t)-\eta \nabla \boldsymbol{L}\left(\mathbf{w}_a(t)\right).$$
>
> Comparing them, omitting higher-order residual terms, we find D-SGD has additional terms as follows,
>
> $$\frac{1}{2} \operatorname{Tr}\left(\boldsymbol{H}\left(\mathbf{w}_a(t)\right) \boldsymbol{\Xi}^*(t)\right) +\mathcal{O}\left(\eta^2 \mathbf{1}\right).$$
>
> The second term $\mathcal{O}\left(\eta^2 \mathbf{1}\right)$ will become negligible compared with the gradient related term $-\eta \nabla\left[\boldsymbol{L}\left(\mathbf{w}_a(t)\right)+\frac{1}{2} \operatorname{Tr}\left(\boldsymbol{H}\left(\mathbf{w}_a(t)\right) \boldsymbol{\Xi}^*(t)\right)\right]=\mathcal{O}\left(\eta \mathbf{1}\right)$ if $\eta$ is sufficiently small. $\frac{1}{2} \operatorname{Tr}\left(\boldsymbol{H}\left(\mathbf{w}_a(t)\right) \boldsymbol{\Xi}^*(t)\right)$ is exactly the regularization term.
>
> > **Q4**: Def.1 is rather weak. Is Definition 1 from a theoretical analysis or just empirical observation?
>
> **A4**: Thanks. We have added a formal definition following your suggestion.
>
> Definition (Linear scaling rule (LSR) (He et al., 2016;  Goyal et al., 2017; Smith et al., 2020; Li et al. 2021)). When multiplying the minibatch size by $\kappa>0$, multiplying the learning rate (LR) also by $\kappa$.
>
> > **Q5**:  From the equation shown in the proof of Theorem 2, there is no batch size. Def. 1 does not imply that here the learning rate can be replaced by the batch size.
>
> **A5**: Thanks. We sincerely note that the noise covariance term in equation (C.2) scales with $1/\text{local batch size}$. The linear scaling rule guarantees $\frac{\eta}{|\mu(t)|}=\mathcal{O}(1).$ Consequently, we obtain $\operatorname{Tr}\left(\Xi_{(t)}\right) = \mathcal{O}\left(|\mu(t)|^2\left(1+\frac{1}{m} \sum_{j=1}^m \frac{1}{\left|\mu_j(t)\right|}\right)\right)$.
>
> > **Q6**: It does not make sense to have a $w^*$ in the non-convex settings. There is no clue that D-SGD can converge to any local minimum in the non-convex settings.
>
> **A6**: We sincerely note that our paper says "if D-SGD is in a local minimum $w^*$. We did not claim that D-SGD necessarily converges to a local minimum. We agree with the reviewer that it is an open problem (very important!) whether D-SGD necessarily converges to a local minimum and can escape saddle points.
>
> > **Q7**: About the reproducibility. No detailed hyperparameters are provided.
>
> **A7**: Thanks. All the implementation settings have been provided on page 8. The code is available at https://github.com/anonymousforrebuttal/anonymous_code.

---

> ### Author Response · Authors · 2022-12-08
> **Sincerely looking forward to your reply**
>
> Dear Reviewer Phej,
>
> Thank you very much for your thorough comments and helpful comments. We are looking forward to your reply. If you have further concerns or requests, please kindly give us a chance to address them in the discussion period. If all your concerns have been resolved, it is much appreciated if you may raise the rating of our work.
>
> Thank you!
>
> The authors

---

### Official Review · Reviewer_ZZMZ · 2022-11-02

**Confidence:** 3
**Correctness:** 3
**Technical Novelty And Significance:** 2
**Empirical Novelty And Significance:** 1
**Recommendation:** 5

**Clarity, Quality, Novelty And Reproducibility:**

Clarity wise is great. The research question is well-motivated and literature review seem solid. I appreciate the author provides proof sketch for friendly explanation.
Quality wise is fair. Please see my questions in Strength And Weaknesses.
Originality wise is good. Although this paper is based on existing technique and framework, the main analysis using Taylor expansion is non-trivial.

**Strength And Weaknesses:**

Strength:
1. First work on the implicit sharpness regularization and escaping efficiency of D-SGD.

Weaknesses:
1. Theoretical justification. Rather than weakness, I would like to raise several questions on the theorem provided. Firstly, in Theorem 1, how does the communication part affect the model update? It seems that the consensus model is merely an average over local models why local models are communicated. I do not see how the later affects the reformulation of Eq. (4). Secondly, in Theorem 2, how does the largest eigenvalue of the Hessian matrix change according to the batch size? Since Theorem 1 suggests the implicit regularization depends on two terms multiplicatively, I think it is necessary to consider the effect of batch size on both of them rather than just one. If $\lambda$ is not constant, it is hard to draw the conclusion that "the sharpness regularization effect of D-SGD is amplified in large-batch settings".
2. Lack of necessary D-SGD examples. While there are at least 4 different topologies and doubly stochastic matrices for D-SGD, I am not sure which one is considered in theorems and experiments. Firstly, Theorem 2 is apparently depending on the spectral gap, which supposes to be different when the topology changes. Can the author provide some characteristics on the scale of $\lambda$? Secondly, it seems that experimental results for sparse topologies can not be found in the current material, while this paper claims "we also conduct experiments on grid-like and static exponential topologies".
3. The empirical evaluation in Figure 1 is not convincing enough. Firstly, the validation accuracy is way below the known accuracy based on the same network structures (e.g.  ResNet-18 can achieve 94% and at least 90%), while the curves seem saturated already. Especially this paper claims that "C-SGD equals to the single-worker SGD with a larger batch size". While I understand the purpose of this paper is not to pursuit SOTA accuracy, but the gap is huge makes me question about the training setup and tuning. Secondly, smaller batch (1024) significantly outperforms larger batch (8192) for both C-SGD and D-SGD. While this paper argues that "these regularization effects are shown to be considerably amplified in large-batch settings" for D-SGD at least and "flat minimizers tend to generalize better than sharp minimizers", I doubt the logic in this paper is complete. Overall, I do not see the empirical evaluation in this paper matches the results in (Zhang et al., 2021), while the later one serves as the motivation of this work.

**Summary Of The Paper:**

This paper provides theoretical and empirical justification of decentralized stochastic gradient descent achieves better validation accuracy than distributed centralized stochastic gradient descent. This paper first proves that D-SGD is equivalent to C-SGD with regularization In Theorem 1. In particular, this regularization term is penalizing both the global model curvature and the consensus distance between the global model and local models. The proof idea is by characterizing the gradient diversity between the global model and local models with second-order Taylor expansion. This paper studies the regularization effect under large batch setting in Theorem 2. In particular, when linear scaling rule is applied, the regularization coefficient of $\lambda_{H,1}$ is cubicly depending on the batch size under the assumption that average local models gradient norm is not decreasing faster than exponent. This paper studies the escaping efficiency of D-SGD in Theorem 3 and Proposition 4. In particular, the grandsum is positive on super-quadratic minima and negative on sub-quadratic minima.

**Summary Of The Review:**

This paper studies an interesting research problem of D-SGD generalization based on recent advances. But before we can trust the insight of this paper, I believe some justification of the results needs to be done. Please see my questions in Strength And Weaknesses.

---

> ### Author Response · Authors · 2022-11-28
> **To reviewer ZZMZ**
>
> Thank you for recognizing the clarity and originality of our paper and we appreciate your constructive comments. The manuscript is carefully revised accordingly. We sincerely hope our responses fully address your questions.
>
> > **Q1**: In Theorem 1, how does the communication part affect the model update?
>
> **A1**: Thanks for your insightful question. The communication part affects the training through the diagonal matrix of ${\Xi}{\scriptstyle (t)}= \frac{1}{m}\sum_{j=1}^{m}(w_{j}{\scriptstyle (t)}-w_{a}{\scriptstyle (t)}){(w_{j}{\scriptstyle (t)}-w_{a}{\scriptstyle (t)})}^T$ (see equation (3)). The effect of communication is incorporated in the last term in the RHS of equation (4) (i.e., the gradient diversity among workers). Specifically, frequent and dense communication (i.e., topologies with large spectral gap (i.e.,  $1-\lambda$) leads to a smaller gradient diversity among the workers and, thus coincides with smaller regularization effect.
>
> > **Q2**: In Theorem 2, how does the largest eigenvalue of the Hessian matrix change according to the batch size? If $\lambda$  is not constant, it is hard to draw the conclusion that "the sharpness regularization effect of D-SGD is amplified in large-batch settings".
>
> **A2**:  Thanks. In theorem 1, $\operatorname{Tr}(\boldsymbol{\Xi}(t))\cdot\lambda_{\boldsymbol{H}\left(\mathbf{w}_a(t)\right), 1}$ is exactly the regularization term.
>
> The second term $\lambda_{\boldsymbol{H}\left(\mathbf{w}_a(t)\right), 1}$ is a sharpness measure. The first term $\operatorname{Tr}(\boldsymbol{\Xi}(t))$ is the "regularization coefficient", characterizing the sharpness regularization’s strength. This term also corresponds to the Lagrangian multiplier in Lasso and Ridge regression. That is, a larger consensus distance $\operatorname{Tr}(\boldsymbol{\Xi}(t))$ will impose a larger implicit penalty on sharpness in decentralized training. Meanwhile, Theorem 2 analyzes the effect of total batch size on the regularization coefficient.
>
> > **Q3**: Which topology is considered in theorems and experiments? Add more experimental results for other sparse topologies.
>
> **A3**: Thanks. Our theorems apply to arbitrary typologies. In the rebuttal, we conducted additional experiments, which currently cover mesh-grid and exponential topologies. The added empirical results are presented at https://github.com/anonymousforrebuttal/anonymous_code.
>
> > **Q4**: The gap between empirical evaluation in Figure 1 and SOTA accuracy.
>
> **A4**: Thanks. This gap is because the training settings in our experiments are not optimized: we did not load pre-train parameters and disable all other training techniques, including dropout, data augmentation, momentum, and weight decay. In the rebuttal, we have re-conducted the experiments with these techniques. We have achieved 94% validation accuracy with ResNet-18 on CIFAR-10. Figure 1 is revised accordingly. The code is available at https://github.com/anonymousforrebuttal/anonymous_code.
>
> > **Q5**: Empirical results in Figure 1 show that the smaller batch (1024) significantly outperforms the larger batch (8192) for both C-SGD and D-SGD and does not support the claim that the regularization effects are amplified in large-batch settings.
>
> **A5**: Thanks. We respectfully note it is **the validation accuracy gap** of C-SGD and D-SGD that characterizes the regularization effects of decentralization. Figure 1 shows that the validation accuracy gap between C-SGD and D-SGD is larger in the larger-batch (8192) setting. This fully supports our claim that the sharpness regularization effect is amplified in large-batch settings, as follows
>
> | Validation acc              |AlexNet               |ResNet-18               |ResNet-34              |
> | ---------------- | -------------------- | -------------------- |-------------------- |
> | D-SGD (1024bs)       | 88.90        |94.94        |95.15        |
> | D-SGD (8192bs)       | 87.47        |94.26        |94.21        |
> | C-SGD (1024bs)       | 87.72        |93.71        |92.14        |
> | C-SGD (8192bs)    | 87.05        |87.28        |72.72        |
>
> Reference:
>
> [1] Sun et al. "Stability and generalization of decentralized stochastic gradient descent." AAAI 2021.
>
> [2] Zhu et al. "Topology-aware generalization of decentralized SGD." ICML 2022.
>
> [3] Keskar et al. "On large-batch training for deep learning: Generalization gap and sharp minima." ICLR 2017.
>
> [4] Hoffer et al. "Train longer, generalize better: closing the generalization gap in large batch training of neural networks." NeurIPS 2017.
>
> [5] Smith et al. "On the generalization benefit of noise in stochastic gradient descent." ICML 2020.

---

### Decision · Program_Chairs · 2023-01-20

**Decision:**

Reject

**Justification For Why Not Higher Score:**

There are numerous major fundamental errors throughout the paper, which appear in definitions, theorems and proofs, including the proof of the main result (Theorem 1). The paper lacks mathematical rigor and basic training in mathematics (reviewers had very detailed reviews and listed numerous errors of this paper).

Since the paper has numerous fundamental errors, I would recommend rejection for this paper.

**Justification For Why Not Lower Score:**

N/A

**Metareview: Summary, Strengths And Weaknesses:**

Summary:

This paper studies decentralized stochastic gradient descent (D-SGD) and compares it to centralized stochastic gradient descent (C-SGD) in terms of generalization performance. The authors present several theoretical results that suggest that D-SGD has an implicit regularization effect, which penalizes the sharpness of the learned minima and the consensus distance between the global and local models. The regularization is amplified in large-batch settings when the linear scaling rule is applied. The authors also study the escaping efficiency of D-SGD, finding that it favors flatter, "super-quadratic" minima. The paper presents experimental results to support the theoretical findings and aims to reconcile conflicting conclusions about the generalization performance of D-SGD.

Strengths:
- The first work on the implicit sharpness regularization and escaping efficiency of D-SGD
- Tackles an interesting and relevant problem (generalization in decentralized settings) and makes an interesting claim about the solutions that D-SGD finds

Weaknesses

- There are numerous major fundamental errors throughout the paper, which appear in definitions, theorems and proofs, including the proof of the main result (Theorem 1). The paper lacks mathematical rigor and basic training in mathematics (reviewers had very detailed reviews and listed numerous errors of this paper).
- Little technical novelty or contribution